# GSM-Symbolic: Understanding the Limitations of Mathematical Reasoning in Large Language Models

**Iman Mirzadeh**[1]     **Keivan Alizadeh**[1]     **Hooman Shahrokhi**[2*]
**Oncel Tuzel**[1]      **Samy Bengio**[1]      **Mehrdad Farajtabar**[1†]
[1]Apple     [2]Washington State University

## Abstract

Recent advancements in Large Language Models (LLMs) have sparked interest in their mathematical reasoning capabilities. While performance on the widely popular GSM8K benchmark has improved, questions remain about whether reported evaluation metrics are reliable, and reasoning abilities of LLMs have advanced. To overcome the limitations of existing evaluations, we introduce GSM-Symbolic, an improved benchmark created from symbolic templates that allow for the generation of a diverse set of questions. GSM-Symbolic enables more controllable evaluations, providing key insights and more reliable metrics for measuring the reasoning capabilities of models. Our findings reveal that LLMs exhibit noticeable variance when responding to different instantiations of the same question. Specifically, the performance of models declines when only the numerical values in the question are altered in the GSM-Symbolic benchmark. Furthermore, we investigate the fragility of mathematical reasoning in these models and demonstrate that their performance significantly deteriorates as the number of clauses in a question increases. We hypothesize that this decline is due to the fact that current LLMs are not capable of genuine logical reasoning; instead, they attempt to replicate the reasoning steps observed in their training data. When we add a single clause that appears relevant to the question, we observe significant performance drops (up to 65%) across all state-of-the-art models, even though the added clause does not contribute to the reasoning chain needed to reach the final answer. Overall, our work provides a more nuanced understanding of LLMs' capabilities and limitations in mathematical reasoning.[1]

## 1 Introduction

Large Language Models (LLMs) have demonstrated remarkable capabilities across various domains, including natural language processing, question answering, and creative tasks (Gunter et al., 2024; OpenAI, 2023; Dubey et al., 2024; Anil et al., 2023; Abdin et al., 2024; Rivière et al., 2024). Their potential to perform complex reasoning tasks, particularly in coding and mathematics, has garnered significant attention from researchers and practitioners.

However, the question of whether current LLMs are genuinely capable of true logical reasoning remains an important research focus. While some studies highlight impressive capabilities, a closer examination reveals substantial limitations. Literature suggests that the reasoning process in LLMs is probabilistic pattern-matching rather than formal reasoning (Jiang et al., 2024). Although LLMs can match more abstract reasoning patterns, they fall short of true logical reasoning. Small changes in input tokens can drastically alter model outputs, indicating a strong token bias and suggesting that these models are highly sensitive and fragile (Jiang et al., 2024; Shi et al., 2023). Additionally, in tasks requiring the correct selection of multiple tokens, the probability of arriving at an accurate answer decreases exponentially with the number of tokens or steps involved, underscoring their inherent unreliability in complex reasoning scenarios (Schaeffer et al., 2023).

---

*Work done during an internship at Apple. †Correspondence to {imirzadeh,farajtabar}@apple.com.
[1]GSM-Symbolic templates and generated data are available at: https://github.com/apple/ml-gsm-symbolic

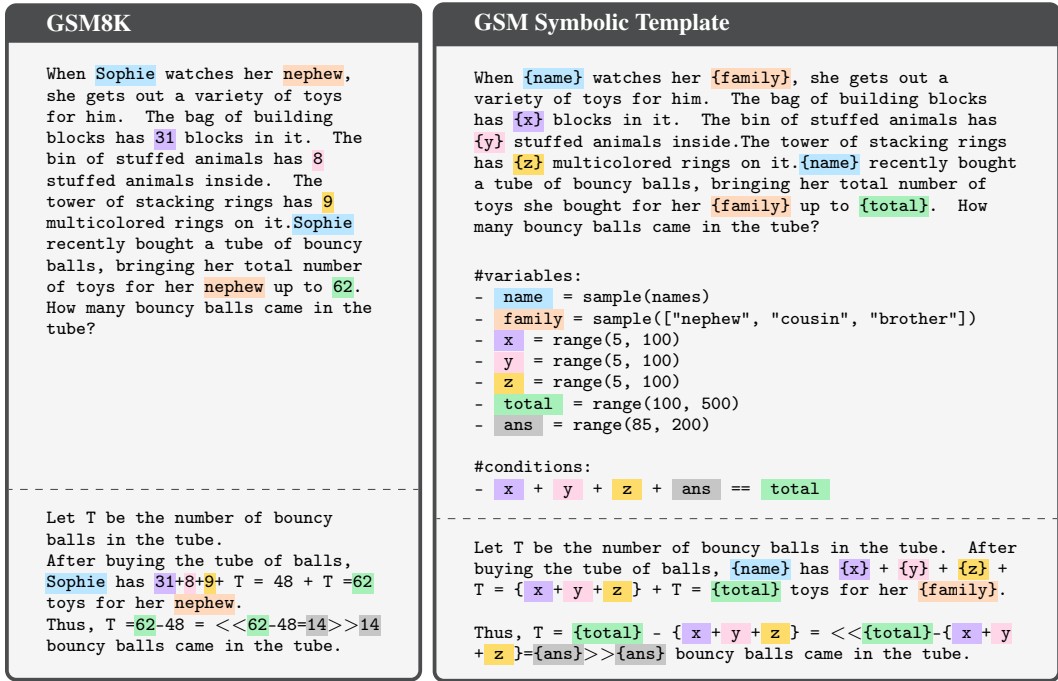

Figure 1: Illustration of the `GSM-Symbolic` template creation process. This dataset serves as a tool to investigate the presumed reasoning capabilities of LLMs, enabling the design of controllable mathematical reasoning evaluations with more reliable metrics. Our results reveal that all state-of-the-art LLMs exhibit significant performance variations, suggesting the fragility or lack of reasoning.

Mathematical reasoning is a crucial cognitive skill that supports problem-solving in numerous scientific and practical applications. Consequently, the ability of large language models (LLMs) to effectively perform mathematical reasoning tasks is key to advancing artificial intelligence and its real-world applications. The GSM8K (Grade School Math 8K) dataset Cobbe et al. (2021) has emerged as a popular benchmark for evaluating the mathematical reasoning capabilities of LLMs. While it includes simple math questions with detailed solutions, making it suitable for techniques like Chain-of-Thought (CoT) prompting, it provides only a single metric on a fixed set of questions. This limitation restricts comprehensive insights into the models' mathematical reasoning. Moreover, the popularity and prevalence of GSM8K can increase the risk of inadvertent data contamination. Finally, the static nature of GSM8K does not allow for controllable experiments to understand model limitations, such as behavior under varied conditions or changes in question aspects and difficulty levels.

To address these limitations, a more versatile and adaptive evaluation framework is needed—one that can generate diverse question variants and adjust complexity levels to better explore the robustness and reasoning abilities of LLMs. This would facilitate a deeper understanding of the strengths and weaknesses of these models in mathematical reasoning tasks. We make the following contributions:

- We introduce `GSM-Symbolic`, an enhanced benchmark that generates diverse variants of `GSM8K` questions using symbolic templates (Sec. 3), as shown in Fig. 1. This enables a more nuanced and reliable evaluation of LLMs' performance across various setups, moving beyond single-point accuracy metrics. Our large-scale study on 25 state-of-the-art open and closed models provides significant insights into LLMs' behavior in mathematical reasoning tasks.

- We question the *reliability* of currently reported results on `GSM8K` and demonstrate that the performance of LLMs can be viewed as a distribution with unwarranted variance across different instantiations of the same question. We show that the performance of all models drops on `GSM-Symbolic` (Sec. 4.1), hinting at potential data contamination.

- We show that LLMs exhibit more robustness to changes in superficial elements like proper names but are very sensitive to changes in numerical values (Sec. 4.2). We show that performance degradation and variance increase as the number of clauses increases, indicating that LLMs' reasoning capabilities struggle with increased complexity (Sec. 4.3).

- Finally, we further question the reasoning abilities of LLMs and introduce the `GSM-NoOp` dataset. By adding seemingly relevant but ultimately irrelevant information to problems, we demonstrate substantial performance drops (up to 65%) across all state-of-the-art models (Sec. 4.4). This reveals a critical flaw in the models' ability to discern relevant information for problem-solving, likely because their reasoning is not formal not formal in the conventional sense and is mostly based on pattern matching. We show that even when provided with multiple examples of the same question or examples containing similar irrelevant information, LLMs struggle to overcome the challenges posed by `GSM-NoOp`. This suggests deeper issues in their reasoning processes that cannot be alleviated by in-context shots and needs further investigation.

Overall, our work provides a comprehensive understanding of the limitations of LLMs in mathematical reasoning. Our results emphasize the need for more reliable evaluation methodologies and further research into the reasoning capabilities of large language models.

## 2 BACKGROUND: REASONING & LANGUAGE MODELS

Logical reasoning is a critical trait of intelligent systems, and building intelligent systems capable of such reasoning has long been an important goal in artificial intelligence (Newell & Simon, 1956; Marcus, 2003; Brachman & Levesque, 2004; Legg & Hutter, 2007; Pearl, 2014; Chollet, 2019). Recent advancements in Large Language Models (LLMs) have demonstrated significant potential across various domains; however, their reasoning abilities remain uncertain and inconsistent. Many works have investigated whether LLMs are truly capable of reasoning by examining how these models solve tasks that require logical reasoning.

We define *logical reasoning* as the *process* by which an agent (a human or a machine) employs logical steps to achieve a *"novel"* goal. The emphasis on "novelty" is crucial because it helps distinguish genuine reasoning from mere memorization of solutions or responses, or mimicking the logical steps previously encountered (Gignac & Szodorai, 2024). This distinction aligns with ideas akin to the Chinese Room Argument (Cole, 2024), underscoring the difference between true comprehension and pattern-matching. Furthermore, this definition connects to several established definitions in AI literature. For instance, the "length-generalization" refers to the ability to apply known logical steps on larger inputs. Moreover, in order to successfully apply such logical steps, an agent often requires other skills such "decomposing" a problem into smaller sub-problems, and "composing" logical steps from these sub-problems in order to solve another problem.

One interesting direction focuses on modeling the computation performed by transformers. For example, parallels have been drawn between components such as attention and feed-forward modules and simple computational primitives (Weiss et al., 2021; Zhou et al., 2024). Delétang et al. (2023) demonstrated that transformers fail to generalize on non-regular tasks and showed that structured memory (e.g., memory tape) is necessary for handling complex tasks. This is related to the effectiveness of Chain-of-Thought (CoT) prompting (Wei et al., 2022) and using scratchpads for LLMs as additional memory for intermediate computations. Moreover, Zubic et al. (2025) show theoretically and empirically that Structured State Space Models (SSMs) and Transformers face fundamental limitations in performing function composition and complex reasoning tasks. Overall, current results suggest that while the transformer architecture has limitations and lacks the required expressiveness for solving problems across several complexity classes, these limitations can be alleviated with additional memory (e.g., scratchpads) (Liu et al., 2024). However, this still requires generating vast amounts of tokens to solve a problem (Peng et al., 2024; OpenAI, 2024). While these works provide insights into the theoretical computational complexity of transformers, in practice, it remains unclear whether these LLMs can perform formal logical reasoning to solve tasks.

There is a considerable body of work suggesting that the reasoning process in LLMs is fragile and not **formal** (Kambhampati, 2024; Valmeekam et al., 2022; 2024; Nezhurina et al., 2024; McCoy et al., 2023; Zhang et al., 2023), even though it appears that these models understand symbols and can work with them to some limited degree (Boix-Adserà et al., 2024). Instead, LLMs likely perform a form of probabilistic **pattern-matching** and searching to find closest seen data during training without proper understanding of concepts. While this process goes beyond naive memorization of words and the models are capable of searching and matching more abstract reasoning steps, it still falls short of true formal reasoning. For instance, Jiang et al. (2024) show, with statistical guarantees, that most LLMs still struggle with logical reasoning due to strong token

bias, where the reasoning output of the model changes when a single token of input changes. This aligns with our results, which indicate that the performance of models on different instances of the same mathematical question can vary greatly from one instance to another. Li et al. (2024b) prove that a single transformer layer learns a one-nearest neighbor, which could explain why the reasoning of models is highly sensitive to input tokens. Schaeffer et al. (2023) argue that when a task requires emitting multiple tokens correctly, the probability of answering correctly decreases exponentially with the number of tokens. Dziri et al. (2023) represent reasoning tasks as computation graphs and find that full computation subgraphs appear much more frequently in training data for correct predictions than incorrect ones. Razeghi et al. (2022) show a correlation between frequency in training and test performance, supporting the pattern matching hypothesis, while Nezhurina et al. (2024) and McCoy et al. (2023) demonstrate the fragility of reasoning in LLMs on simple tasks.

Our work builds upon these findings by introducing `GSM-Symbolic`, an improved benchmark using symbolic templates to generate diverse question variants. This allows us to study mathematical reasoning ability beyond a single performance metric. By evaluating performance on different instantiations and difficulty levels, we draw a comprehensive picture of LLMs' reasoning capabilities. Related to this direction, Stolfo et al. (2023) evaluate LLMs on math word problems by measuring sensitivity and robustness to different input interventions and report that models often rely on surface-level cues rather than genuine reasoning, while Hong et al. (2024) developed an ontology and used it to perturb the GSM8K dataset with the aid of LLMs, and Srivastava et al. (2024) purposes functional variant of MATH dataset. Our findings in this work support the hypothesis that current LLMs are not capable of performing formal mathematical reasoning and pave the way for further research.

## 3 GSM-SYMBOLIC

The `GSM8K` dataset (Cobbe et al., 2021) includes over 8000 grade school math questions and answers, divided into 7473 training and 1319 test examples. As shown in Fig. 1, the questions are relatively simple, requiring knowledge of only the four main arithmetic operations. However, since `GSM8K` is a single, popular test set, there is a risk of data contamination, and performance may change significantly with minor modifications to the questions. These limitations have led to efforts to generate new datasets and variants. `iGSM` (Ye et al., 2024) is a math dataset created through a synthetic pipeline that captures parameter dependencies in a hierarchical and graph structure. `GSM-IC` (Shi et al., 2023) shows that irrelevant context can impair LLM performance, focusing on prompting techniques. Our work, however, suggests a more fundamental issue: LLMs struggle even when given multiple shots of the same question, indicating deeper challenges in problem-solving that cannot be resolved with few-shot prompting or fine-tuning on unseen distractions or variations of the same or different difficulty levels. `GSM-Plus` (Li et al., 2024a) introduces variants of GSM8K questions but lacks symbolic templates and has a fixed size and difficulty. `GSM1K` (Zhang et al., 2024) mirrors the style and complexity of `GSM8K` to identify systematic overfitting in existing models, but it is not publicly available for researchers.

While the mentioned benchmarks offer a single performance metric on a fixed number of questions, we argue that viewing LLM performance as a distribution across various problem instances provides deeper insights. The design of `GSM-Symbolic` enables the generation of numerous instances and allows for finer control over question difficulty. We believe our paper contributes to this direction by offering a reliable evaluation framework that underscores the importance of generating multiple instances to assess LLMs' mathematical capabilities and their robustness to diverse problem difficulties and augmentations.

### 3.1 TEMPLATE GENERATION

Given a specific example from the test set of `GSM8K`, we create parsable templates as shown in Fig. 1 (right). The annotation process involves identifying variables, their domains, and necessary conditions to ensure the correctness of both the question and the answer. For instance, since the questions are grade-school level, a common condition is divisibility to ensure the answer is a whole number. We use common proper names (e.g., persons, foods, currencies) to streamline creation. After creating the templates, we apply several automated checks to ensure the annotation process is correct. For example, we verify that none of the original variable values appear in the template.

We also check that the original values satisfy all conditions and that the final answer matches the original question's answer. Once data are generated, 10 random samples per template are reviewed manually. As a final automated check, after evaluating all models, we verify that at least two models answer each question correctly; otherwise, the question is reviewed manually again.

When constructing symbolic templates, we deliberately select numerical ranges that closely align with those in the original `GSM8K` test set. This decision reflects our focus on assessing logical reasoning capabilities rather than arithmetic skills. Our analysis in Appendix A.6 confirms that the expanded ranges remain within boundaries where models maintain their arithmetic accuracy.

## 3.2 Experimental Setup

While we provide further details on our experimental setup and evaluation in the Appendix, we briefly review the important aspects here:

**Models.** Throughout this work, we report on more than 20 open models of various sizes, ranging from 2B to 27B. Additionally, we include state-of-the-art closed models such as GPT-4o-mini, GPT-4o, o1-mini, and o1-preview. To conserve space, we present results for a few selected models in each experiment, but the full results for all models are available in Tab. 1 of the Appendix A.2.

**Evaluation Setup** Overall, for this work, we conducted nearly 500 total evaluations on various setups. To this end, we maintained a manageable dataset size by using 100 templates and generating 50 samples per template, resulting in 5000 total examples for each benchmark. Therefore, we have 50 datasets of 100 examples each, where each example is a mutation of one of the original 100 examples from `GSM8K`. Unless stated otherwise, we follow a common evaluation setup on GSM8K and other math benchmarks that includes Chain-of-Thought (CoT) prompting with 8-shots with greedy decoding. However, we note that in our preliminary experiments, the number of shots did not significantly change the performance and conclusions. We provide our prompt template in Fig. 9.

## 4 Experiments & Results

In this section, we present our main results and postpone complementary findings to the Appendix. We begin our experiments by addressing an important question regarding the reliability of current reported metrics on `GSM8K`. By studying the *distribution* of performance on `GSM-Symbolic`, we demonstrate notable performance variation. More importantly, we observe that the performance of models drops on `GSM-Symbolic` (Sec. 4.1).

Next, we investigate the fragility of reasoning in LLMs by comparing performance distributions when only proper names are changed versus when values and numbers are altered. Our findings indicate that the original `GSM8K` performance of models is much closer to the performance distribution when only names are changed. However, performance drops more significantly when values are changed, with this trend continuing as both changes are applied simultaneously (Sec. 4.2). We then examine the impact of question difficulty, as indicated by the number of clauses added to or removed from the questions. Our results show that as the number of clauses increases, average performance drops, and the variance in performance increases consistently across all models (Sec. 4.3).

Finally, in Sec. 4.4, we tackle a more fundamental question: whether the models truly understand the mathematical concepts. We show that, likely due to potential pattern matching and the fact that the training distribution of models included only necessary information for solving questions, adding seemingly relevant clauses to the question that do not impact the reasoning process required to solve it significantly drops the performance of all models.

## 4.1 How Reliable Are the Current GSM8K Results?

As our first experiment, we evaluate the performance of several state-of-the-art models on `GSM-Symbolic`. The number of samples and difficulty can be adjusted by modifying variable domains, as we will see in subsequent sections. Fig. 2 shows the empirical distribution of the performance of models on `GSM-Symbolic` computed on these 50 datasets. As shown, all models exhibit a non-negligible variance across different sets. For instance, for the Gemma2-9B, the gap between the worst performance and the best performance is more than 12%, while for Phi-3.5-mini, this gap

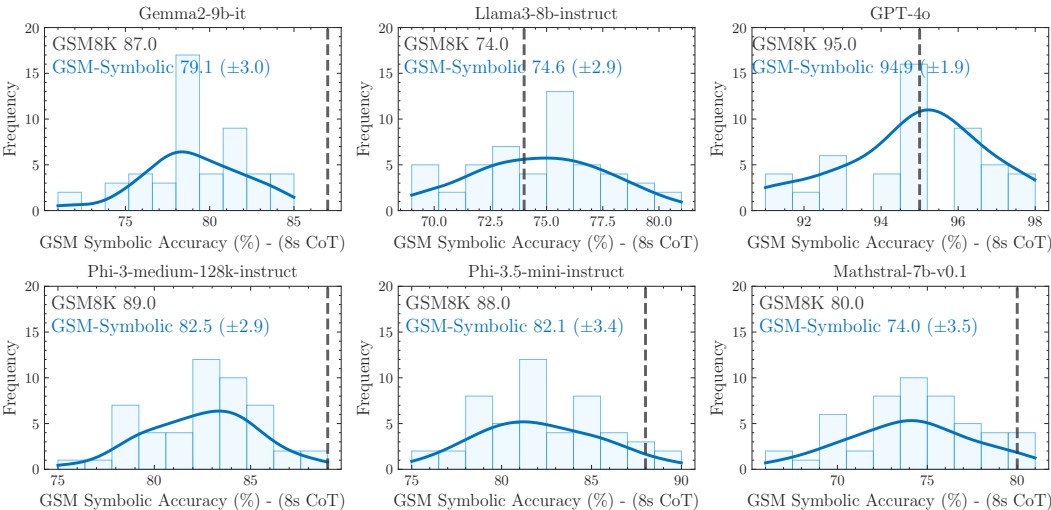

Figure 2: 8-shot CoT performance across 50 sets generated from `GSM-Symbolic` templates. All state-of-the-art models exhibit notable variance in accuracy. It is interesting that for the majority of the models, the performance on `GSM8K` (represented by dashed line) falls on the right side of the distribution, which statistically speaking, should have a very low likelihood.

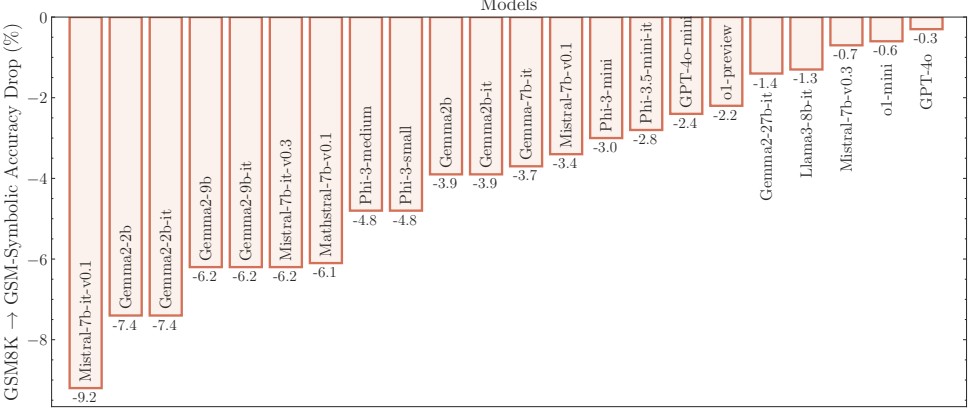

Figure 3: The performance of all state-of-the-art models on `GSM-Symbolic` drops compared to `GSM8K`. Later, we investigate the factors that impact the performance drops in more depth.

is around 15%. It is interesting that this variation even exists, as the only differences across different instances of each question are the changes in names and values, while the overall reasoning steps needed to solve a question remain the same.

Another noteworthy observation is that the performance (represented by the dashed line in Fig. 2) on the original questions from the 100 examples of `GSM8K` used as templates is often more than one standard deviation away from the center of the `GSM-Symbolic` performance distribution, frequently on the *right side* of the distribution (this holds for 21 out of 25 models). One explanation for this could be data contamination, where some of the test examples from `GSM8K` inadvertently ended up in the training set of these models, leading to an optimistic bias in performance. Fig. 3 shows the performance drop from `GSM8K` to `GSM-Symbolic` for several models. We can see that for models such as Gemma2-9B, Phi-3, Phi-3.5, and Mathstral-7B, the dashed line in Fig. 2 lies on the right side, and the drop in performance is higher than for models such as Llama3-8b and GPT-4o, where the performance on `GSM8K` is close to the center of the `GSM-Symbolic` distribution and the drop in performance is negligible. In Appendix A.3, we present further results to support this claim for other models such as Phi-2 and Mistral-7B. These results lead us to investigate the fragility of the reasoning abilities of LLMs in the next section.

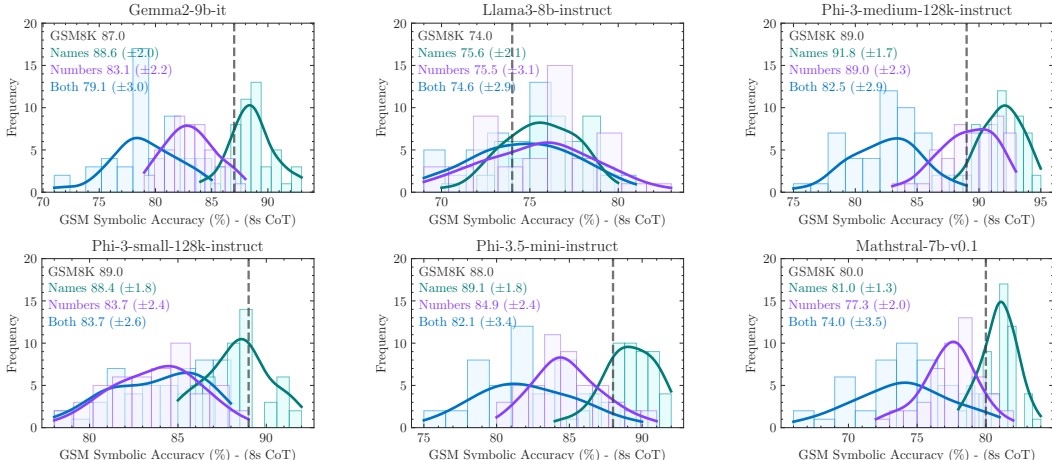

Figure 4: How sensitive are LLMs when we change only names, only proper numbers, or both names and numbers? Overall, models have noticeable performance variation even if we only change names, but even more when we change numbers or combine these changes.

## 4.2 How Fragile is Mathematical Reasoning in Large Language Models?

In the previous sub-section, we observed high performance variation across different sets generated from the same templates, along with a performance degradation compared to the original GSM8K accuracy. This suggests that the perceived reasoning process of language models may not be formal and is hence susceptible to changes. One explanation is that these models attempt to perform a kind of in-distribution pattern-matching, aligning given questions and solution steps with similar ones seen in the training data. As no formal reasoning is involved in this process, it could lead to high variance across different instances of the same question. In this sub-section and the next one, we investigate these observations further and we show that several factors contribute to the performance variation of the models. First, we investigate the impact of the *type* of change to understand the difference between changing names (e.g., person names, places, foods, currencies, etc.) versus changing numbers (i.e., the values of variables).

Figure 4 demonstrates that while performance variation persists, the variance is lower when changing names compared to numbers. Notably, the original GSM8K accuracy of models is now much closer to the center of the changed proper names distribution, in contrast to changed numbers or both. Furthermore, a gradual shift in the means of distributions from right to left, along with an increase in variance, is evident across almost all models. It is both striking and concerning that such performance variance exists when only changing proper names, as this level of variability would not be expected from a grade-school student with genuine mathematical understanding.

From the results in this section, we observe that by increasing the *difficulty* of changes (from names to numbers), the performance drops and the variance increases, overall suggesting that the reasoning capabilities of state-of-the-art LLMs are fragile for the aforementioned reasons. Assuming that LLMs are not performing formal reasoning, how important is the question *difficulty* on the distribution of performance? In the next section, we study this question further.

## 4.3 How Does Question Difficulty Affect Performance Distribution?

The results in the previous subsection motivate us to study the impact of question *difficulty* on the mean and variance of the performance distribution. To this end, we generate several new templates from the GSM-Symb, as illustrated in Fig. 5. First, by removing one clause, we obtain GSM-Symbolic-Minus-1 or GSM-M1 for short. Similarly, we can add one or two clauses

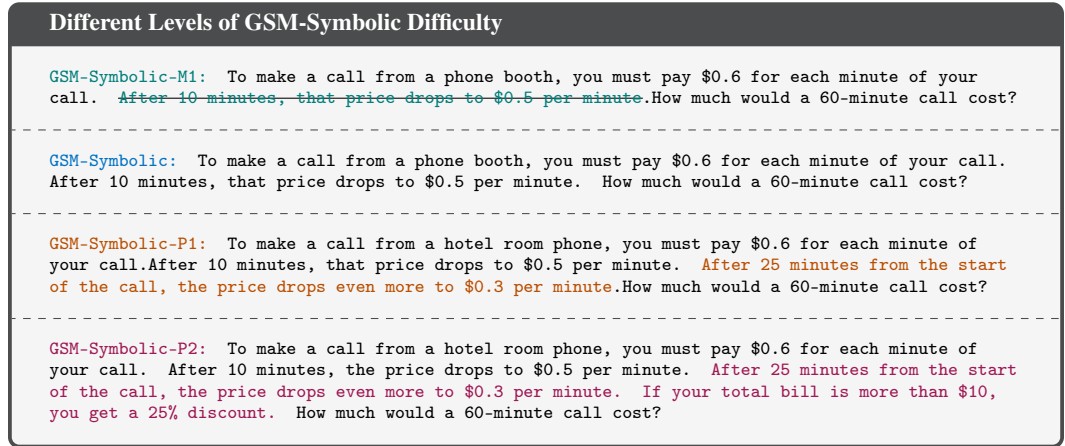

Figure 5: Modifying the difficulty level of GSM-Symbolic by modifying the number of clauses.

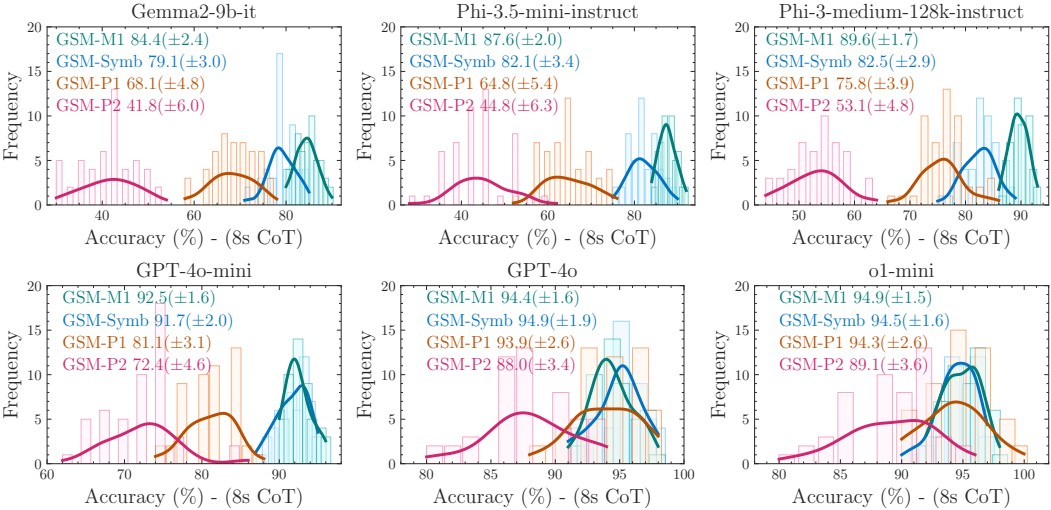

Figure 6: The impact of increasing the number of clauses on performance: As the difficulty increases from `GSM-M1` → `GSM-Symb` → `GSM-P1` → `GSM-P2`, the distribution of performance shifts to the left (i.e., accuracy decreases), and the variance increases.

to the questions to increase the difficulty, resulting in `GSM-Symbolic-Plus-1` (`GSM-P1`) and `GSM-Symbolic-Plus-2` (`GSM-P2`), respectively[2].

As shown in Fig. 6, the trend of the evolution of the performance distribution is very consistent across all models: as the difficulty increases, the performance decreases and the variance increases. Note that overall, the *rate of accuracy drop* also increases as the difficulty increases. This is in line with the hypothesis that models are not performing formal reasoning, as the number of required reasoning steps increases linearly, but the rate of drop seems to be faster. Moreover, considering the pattern-matching hypothesis, the increase in variance suggests that searching and pattern-matching become significantly harder for models as the difficulty increases.

---

[2]Note that adding or removing a clause does not necessarily correspond to increasing or decreasing the number of required reasoning steps by exactly one. However, our main focus in this section is to understand the *evolution* of the performance distribution rather than the precise performance numbers.

---

**GSM-NoOp**

Oliver picks 44 kiwis on Friday. Then he picks 58 kiwis on Saturday. On Sunday, he picks double the number of kiwis he did on Friday, `but five of them were a bit smaller than average`. How many kiwis does Oliver have?

- - - - - - - - - - - - - - - - - - - - - - - - - - - - - - - - - - - - - - - - - - - - - - - - - - - - - - -

`o1-mini`: Oliver picks kiwis over three days as follows:
- Friday: 44 kiwis
- Saturday: 58 kiwis
- Sunday: Double the number he picked on Friday, which is $2 \times 44 = 88$ kiwis
However, on Sunday, 5 of these kiwis were smaller than average. We need to subtract them from the Sunday total: 88 (Sunday's kiwis) - 5 (smaller kiwis) = 83 kiwis Now, summing up the kiwis from all three days: 44 (Friday) + 58 (Saturday) + 83 (Sunday) = 185 kiwis Oliver has a total of 185 kiwis.

- - - - - - - - - - - - - - - - - - - - - - - - - - - - - - - - - - - - - - - - - - - - - - - - - - - - - - -

`Llama3-8B`: Let's break it down step by step:
Oliver picks 44 kiwis on Friday. He picks 58 kiwis on Saturday. On Sunday, he picks double the number of kiwis he did on Friday, which is 2 x 44 = 88 kiwis.
Five of the kiwis picked on Sunday are a bit smaller than average, so we subtract 5 from the total number of kiwis picked on Sunday: 88 - 5 = 83 kiwis. Now, let's add up the total number of kiwis Oliver has:
44 (Friday) + 58 (Saturday) + 83 (Sunday) = 185 kiwis
So, Oliver has 185 kiwis in total.

Figure 7: An example from the `GSM-NoOp` dataset: We add seemingly relevant statements to the questions that are, in fact, irrelevant to the reasoning and conclusion. However, the majority of models fail to ignore these statements and blindly convert them into operations, leading to mistakes.

## 4.4 CAN LLMS REALLY UNDERSTAND MATHEMATICAL CONCEPTS?

In the previous sections, we studied the impact of *type of change* and *difficulty* on the performance distribution. In this section, we demonstrate that models are susceptible to catastrophic performance drops on instances not part of the training distribution, potentially due to their reliance on in-distribution pattern-matching. We introduce `GSM-NoOp`, a dataset designed to challenge the reasoning capabilities of language models. To create the templates, we add seemingly relevant but ultimately inconsequential statements to `GSM-Symbolic` templates. Since these statements carry no operational significance, we refer to them as *"No-Op"*.

Fig. 7 illustrates an example from `GSM-NoOp`. An interesting observation is that models tend to blindly subtract the number of smaller fruits, potentially because their training datasets included similar examples that required conversion to subtraction operations. In the Appendix, we include additional failure cases from `GSM-NoOp`. Overall, we find that models tend to convert statements to operations without truly understanding their meaning. For instance, a common case we observe is that models interpret statements about "discount" as "multiplication", regardless of the context. This raises the question of whether these models have truly understood the mathematical concepts well enough. Consequently, as shown in Fig. 8a, there is a catastrophic performance decline across all tested models, with the Phi-3-mini model experiencing over a 65% drop, and even stronger models such as o1-preview showing significant declines.

To better understand this performance drop, we conducted another experiment. While our previous evaluations on `GSM-P2` used the original 8-shots of `GSM8K`, here we explore two new scenarios where we change the source of the 8-shots. We report the results in Figures 8b and 8c.

**NoOp-Symb** (Using `GSM-Symbolic` shots of the *same* question): During evaluation, we include 8 different shots of the *same* question coming from `GSM-Symbolic`. Hence, each shot provides the required reasoning steps. The target question from `GSM-NoOp` then presents yet another variation of the same question that is different only in values and the added clause that is inconsequential. This setup should simplify the task by making it clear that the extra information in the target question is irrelevant. However, as shown in Fig. 8b, the performance remains within the standard deviation, even with 8 shots of the same question providing the reasoning chain. Interestingly, Fig. 8c shows that some models can perform significantly better, even though they don't perform nearly as well on `GSM8K` and `GSM-Symbolic`. We believe this is a very notable observation.

**NoOp-NoOp** (Using `GSM-NoOp` shots of *different* questions): Here, we provide 8 shots chosen randomly from different questions of `GSM-NoOp` in the context. These questions share the common

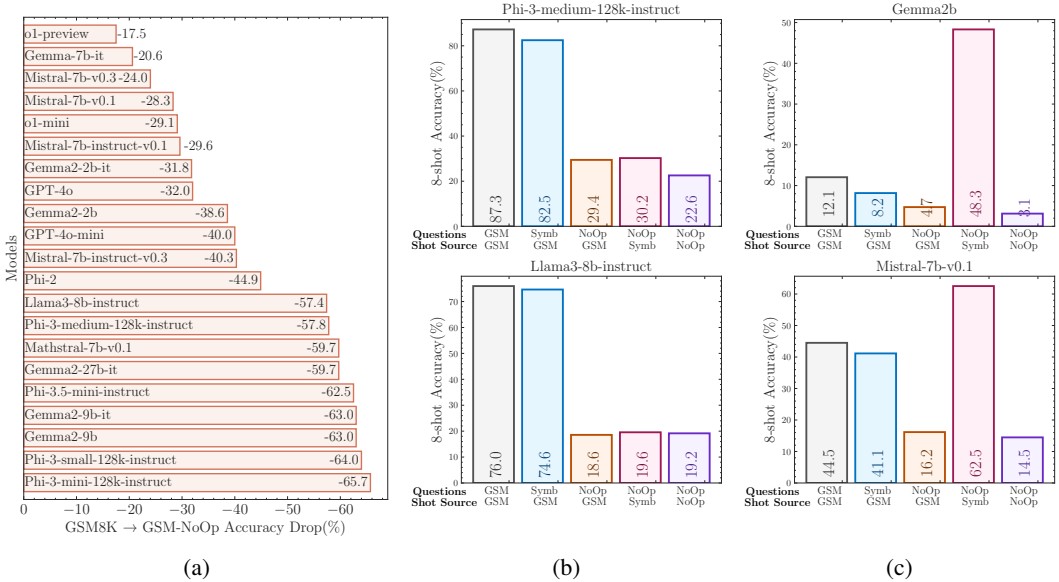

Figure 8: **(a)** The performance of models drops significantly on `GSM-NoOp`, with more recent models experiencing a greater decline than older ones. **(b)** As previously demonstrated, performance on `GSM-Symbolic` is very close to that on `GSM8K`. However, on `GSM-NoOp`, the significant drop in performance cannot be recovered, even when using the exact same question's variation as shots (`NoOp-Symb`) or when using different questions with different `GSM-NoOp` that contain No-Op operations (`NoOp-NoOp`) as shots. **(c)** Notably, some models that perform significantly worse than those in (b) on `GSM8K` and `GSM-Symbolic` show much better performance on `NoOp-Symb`.

fact that the correct answer should ignore the No-Op statement. We observe that for the Llama-3-8B model, the performance remains the same compared to the original No-Op model, while for the Phi-3 model, performance slightly decreases.

## 5 CONCLUSION

In this work, we have investigated the reasoning capabilities of large language models (LLMs) and the limitations of current evaluations on GSM8K. We introduced GSM-Symbolic, a novel benchmark with multiple variants designed to provide deeper insights into the mathematical reasoning abilities of LLMs. Our extensive study reveals significant performance variability across different instantiations of the same question, challenging the reliability of current GSM8K results that rely on single-point accuracy metrics. We found that while LLMs exhibit some robustness to changes in proper names, they are more sensitive to variations in numerical values. We have also observed the performance of LLMs deteriorating as question complexity increases.

The introduction of GSM-NoOp exposes a critical flaw in LLMs' ability to genuinely understand mathematical concepts and discern relevant information for problem-solving. Adding seemingly relevant but ultimately inconsequential information to the logical reasoning of the problem led to substantial performance drops across models. Importantly, we demonstrate that LLMs struggle even when provided with multiple examples of the same question or examples containing similar irrelevant information. This suggests deeper issues in their reasoning processes that cannot be easily mitigated through few-shot learning or fine-tuning.

Ultimately, our work underscores significant limitations in LLMs' ability to perform genuine mathematical reasoning. The LLMs' high performance variance on different instances of the same question, their significant drop in performance with a slight increase in difficulty, and their sensitivity to inconsequential information indicate that their reasoning is fragile and may be more akin to sophisticated pattern matching rather than true logical reasoning. We believe further research is needed to develop AI systems capable of formal reasoning, moving beyond probabilistic pattern matching to achieve more robust and generalizable problem-solving skills. This remains a critical challenge for the field as we strive to create systems with human-like cognitive abilities or general intelligence.

ACKNOWLEDGMENTS

The authors would like to thank Max Horton, Fartash Faghri, Moin Nabi, Devi Krishna, and Arsalan Farooq for the valuable feedback and support.

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

# A  APPENDIX

In this appendix, we provide additional details to the main text, including:

- A.1: Detailed experimental setups, including the prompt template.
- A.2: Full results on `GSM8K`, `GSM-Symbolic`, and their variants.
- A.3: Additional results for the distributional performance of several models, similar to the results from Sec. 4.1 in the main text.
- A.4: Additional results for Sec. 4.3, where we studied the impact of question difficulty. We show that fine-tuning on easier tasks does not necessarily improve performance on more difficult tasks.
- A.5: A more comprehensive discussion and analysis of performance for OpenAI o1-mini and o1-preview models.

## A.1  DETAILED EXPERIMENTAL SETUP

In this work, all reported evaluations results use 8-shots with chain-of-thought prompting. We use the following prompt format:

```
Evaluation Prompt Format

// preamble or system instruction
As an expert problem solver, solve step by step the following mathematical questions.

// shot-1
Q: {{question}}
A: Let's think step by step.  {{solution}}.  The final answer is {{final answer}}.
.
.
.
// shot 8
Q: {{question}}
A: Let's think step by step.  {{solution}}.  The final answer is {{final answer}}.

// target question
Q: {{question}}
A: Let's think step by step.
```

Figure 9: The prompt format used for evaluations.

Except for the last experiment in Sec. 4.4, we use the original 8 shots from `GSM8K`. In addition, we allow the models to generate until either their context size limit is reached, they generate one of the end-of-response tokens such as '``' or '`<|endoftext|>`', or they finish answering the current question and move on to generating the next question, indicated by another '`Q:`' generation.

Finally, we note that in all experiments we use greedy decoding to generate responses from models, with one exception: currently, the available APIs for "o1-mini" and "o1-preview" models do not allow controlling the decoding strategy, and it seems that at the time of writing, these models do not perform greedy decoding, as responses to the same prompt change.

## A.2  FULL RESULTS

In Tab. 1, we present the comprehensive performance results of various models, including Gemma (Mesnard et al., 2024), Gemma2 (Rivière et al., 2024), Phi (Abdin et al., 2024), Mistral (Jiang et al., 2023), Llama3 (Dubey et al., 2024), GPT-4o (OpenAI, 2023), and the o1 (OpenAI, 2024) series, on `GSM8K` and its different variants, `GSM-Symbolic`.

We report two sets of results for `GSM8K`: the first column indicates the accuracy on the *full* test set of `GSM8K` (comprising 1,319 examples), while the second column shows the accuracy on a subset of 100 questions from the `GSM8K` test set, which we randomly selected to generate `GSM-Symbolic`

templates. It is noteworthy that the performance levels across both sets are very similar, with no significant differences observed.

Table 1: Full 8-shot results of all models on `GSM8K` and different variants of `GSM-Symbolic`.

| Model | GSM8K (Full) | GSM8K (100) | Symbolic-M1 | Symbolic | Symbolic-P1 | Symbolic-P2 | Symbolic-NoOp |
|---|---|---|---|---|---|---|---|
| Gemma2b | 12.1 | 11.0 | 24.5 (± 3.85) | 8.2 (± 2.21) | 3.6 (± 2.13) | 1.5 (± 1.63) | 4.7 (± 1.99) |
| Gemma2b-it | 12.1 | 11.0 | 16.2 (± 3.28) | 8.2 (± 2.21) | 1.5 (± 1.49) | 1.5 (± 1.63) | 4.1 (± 2.48) |
| Gemma-7b | 53.8 | 50.0 | 34.1 (± 4.41) | 25.6 (± 3.25) | 26.0 (± 5.30) | 3.1 (± 1.92) | 8.7 (± 2.71) |
| Gemma-7b-it | 29.3 | 33.0 | 34.1 (± 4.41) | 25.6 (± 3.25) | 3.1 (± 1.92) | 3.1 (± 1.92) | 8.7 (± 2.71) |
| Gemma2-2b | 47.5 | 46.0 | 57.2 (± 3.40) | 40.1 (± 3.04) | 19.5 (± 3.89) | 1.3 (± 1.37) | 8.8 (± 4.12) |
| Gemma2-2b-it | 47.5 | 46.0 | 57.2 (± 3.40) | 40.1 (± 3.04) | 19.5 (± 3.89) | 4.5 (± 1.94) | 15.7 (± 3.97) |
| Gemma2-9b | 85.3 | 87.0 | 71.2 (± 2.81) | 79.1 (± 2.99) | 44.0 (± 5.69) | 41.8 (± 6.00) | 22.3 (± 5.11) |
| Gemma2-9b-it | 85.3 | 87.0 | 84.4 (± 2.36) | 79.1 (± 2.99) | 68.1 (± 4.77) | 41.8 (± 6.00) | 22.3 (± 5.11) |
| Gemma2-27b-it | 89.7 | 92.0 | 90.2 (± 1.86) | 88.3 (± 2.56) | 80.7 (± 4.07) | 63.4 (± 4.14) | 30.0 (± 3.39) |
| Phi-2 | 56.0 | 53.0 | 53.0 (± 3.10) | 41.4 (± 3.56) | 23.3 (± 4.07) | 8.9 (± 3.33) | 11.2 (± 3.51) |
| Phi-3-mini-128k-instruct | 83.7 | 85.0 | 85.9 (± 2.44) | 80.7 (± 2.94) | 63.4 (± 5.63) | 37.5 (± 5.76) | 18.0 (± 3.83) |
| Phi-3-small-128k-instruct | 88.5 | 89.0 | 86.4 (± 1.95) | 83.7 (± 2.65) | 72.0 (± 3.65) | 50.7 (± 4.99) | 24.5 (± 4.81) |
| Phi-3-medium-128k-instruct | 87.3 | 89.0 | 89.6 (± 1.65) | 82.5 (± 2.86) | 75.8 (± 3.89) | 53.1 (± 4.80) | 29.4 (± 4.18) |
| Phi-3.5-mini-instruct | 84.9 | 88.0 | 87.6 (± 1.98) | 82.1 (± 3.38) | 64.8 (± 5.43) | 44.8 (± 6.32) | 22.4 (± 4.03) |
| Mistral-7b-v0.1 | 44.5 | 48.0 | 55.4 (± 3.18) | 41.1 (± 3.36) | 17.4 (± 4.82) | 5.5 (± 2.55) | 16.2 (± 4.43) |
| Mistral-7b-instruct-v0.1 | 39.7 | 42.0 | 44.9 (± 4.29) | 30.5 (± 3.47) | 13.1 (± 3.51) | 4.0 (± 2.24) | 10.1 (± 3.42) |
| Mistral-7b-v0.3 | 40.6 | 44.0 | 54.0 (± 2.95) | 40.0 (± 4.43) | 15.6 (± 4.02) | 3.9 (± 2.31) | 16.7 (± 4.26) |
| Mistral-7b-instruct-v0.3 | 56.2 | 56.0 | 62.3 (± 2.68) | 50.0 (± 3.49) | 24.5 (± 4.34) | 10.8 (± 3.60) | 15.9 (± 4.44) |
| Mathstral-7b-v0.1 | 80.1 | 80.0 | 82.9 (± 2.87) | 74.0 (± 3.49) | 57.4 (± 5.20) | 35.5 (± 5.07) | 20.4 (± 3.58) |
| Llama3-8b | 55.8 | 61.0 | 79.5 (± 3.62) | 74.6 (± 2.94) | 53.8 (± 4.54) | 12.3 (± 3.43) | 18.6 (± 3.86) |
| Llama3-8b-instruct | 76.0 | 74.0 | 79.5 (± 3.62) | 74.6 (± 2.94) | 53.8 (± 4.54) | 28.3 (± 4.37) | 18.6 (± 3.86) |
| GPT-4o-mini | 94.2 | 95.0 | 92.5 (± 1.63) | 91.7 (± 2.02) | 81.1 (± 3.05) | 72.4 (± 4.57) | 54.1 (± 3.85) |
| GPT-4o | 95.2 | 95.0 | 94.4 (± 1.62) | 94.9 (± 1.87) | 93.9 (± 2.59) | 88.0 (± 3.43) | 63.1 (± 4.53) |
| o1-mini | 95.1 | 93.0 | 94.9 (± 1.49) | 94.5 (± 1.58) | 94.3 (± 2.57) | 89.1 (± 3.56) | 66.0 (± 4.60) |
| o1-preview | 94.9 | 96.0 | 93.6 (± 1.68) | 92.7 (± 1.82) | 95.4 (± 1.72) | 94.0 (± 2.38) | 77.4 (± 3.84) |

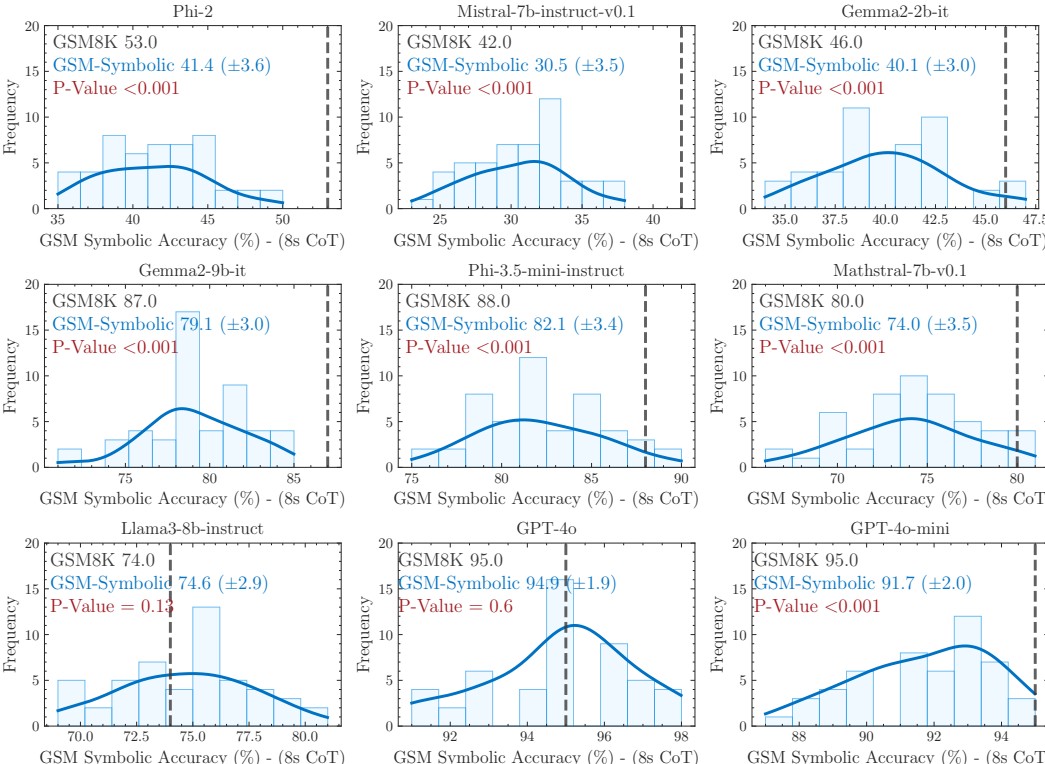

Figure 10: Additional results on performance variation on `GSM-Symbolic`.

### A.3 ADDITIONAL RESULTS ON GSM-SYMBOLIC PERFORMANCE DISTRIBUTIONS

In section 4.1, we have presented results for several models in Fig. 2. Here, we provide additional results showing the performance on `GSM-Symbolic` for other models also have high variance. Moreover, these models correspond to highest drop.

Another important question regarding our results is measuring the statistical significance[3]. In this work, we calculate statistical significance using the one-sample t-test to determine whether the 50 different performance results on `GSM-Symbolic` differ from the original `GSM8K` score (i.e., the null hypothesis)[4]. As we can see in Fig. 10, for an overwhelming majority of models (except Llama3-8B and GPT-4o), the results are statistically significant.

### A.4 ABLATION: DOES FINE-TUNING ON EASIER TASKS HELP WITH MORE DIFFICULT TASKS?

In Sec. 4.3, we observed that the performance on `GSM-P2` is significantly lower than the performance on `GSM-P1`. We also argued that it is unlikely that additional fine-tuning or including shots from `GSM-P1` would be beneficial. Here, in Fig. 11a, we show that including shots from `GSM-P1` does not improve performance compared to the results where shots come solely from `GSM8K`.

Moreover, in Fig. 11b, we demonstrate that fine-tuning Phi-3.5 on `GSM-P1` slightly improves performance on `GSM-P1` while decreasing performance on `GSM-P2`. We have used a set of 50 templates from `GSM-P1`, separate from the test templates, and generated 10000 examples for finetuning training set.

Overall, while this direction warrants further research, current results suggest that scaling training data will not be helpful in improving the reasoning capabilities of language models.

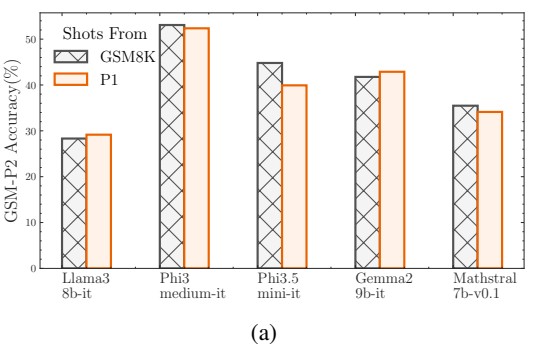
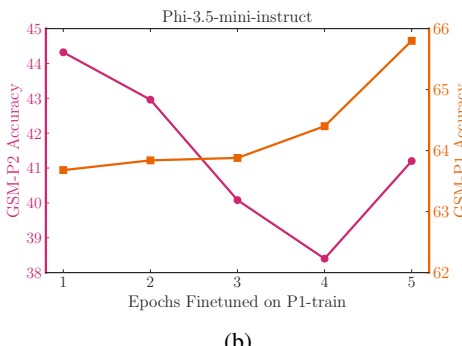

(a)                                                                (b)

Figure 11: Using in-context shots or finetuning on `GSM-P1` does not improve performance on `GSM-P2`: (a) Compared to the case where 8 shots come from `GSM8K`, when we include shots from `GSM-P1` the performance on `GSM-P2` does not improve. (b) Finetuning on `GSM-P1` can improve performance on `GSM-P1` but not on `GSM-P2`.

### A.5 RESULTS ON O1-PREVIEW AND O1-MINI

The recently released o1-preview and o1-mini models (OpenAI, 2024) have demonstrated strong performance on various reasoning and knowledge-based benchmarks. As observed in Tab. 1, the mean of their performance distribution is significantly higher than that of other open models.

---

[3]This is a very complicated topic, as the notion of statistical significance depends on many assumptions about the properties of data. For instance, one could view the overall performance of models as a Bernoulli trial, with model accuracy representing the probability of success. However, this requires an i.i.d. assumption on the questions and the model accuracy for each question, which may not necessarily hold and needs further investigation.

[4]In general, the t-test assumes normality, which may not necessarily hold for evaluation results. However, using a standard normality test based on the skewness and kurtosis of samples, we have verified that all distributions in Fig. 10 pass the normality test (p-value of the normality test = 0.1).

In Fig. 12 (top), we illustrate that both models exhibit non-negligible performance variation. When the difficulty level is altered, o1-mini follows a similar pattern to other open models: as the difficulty increases, performance decreases and variance increases.

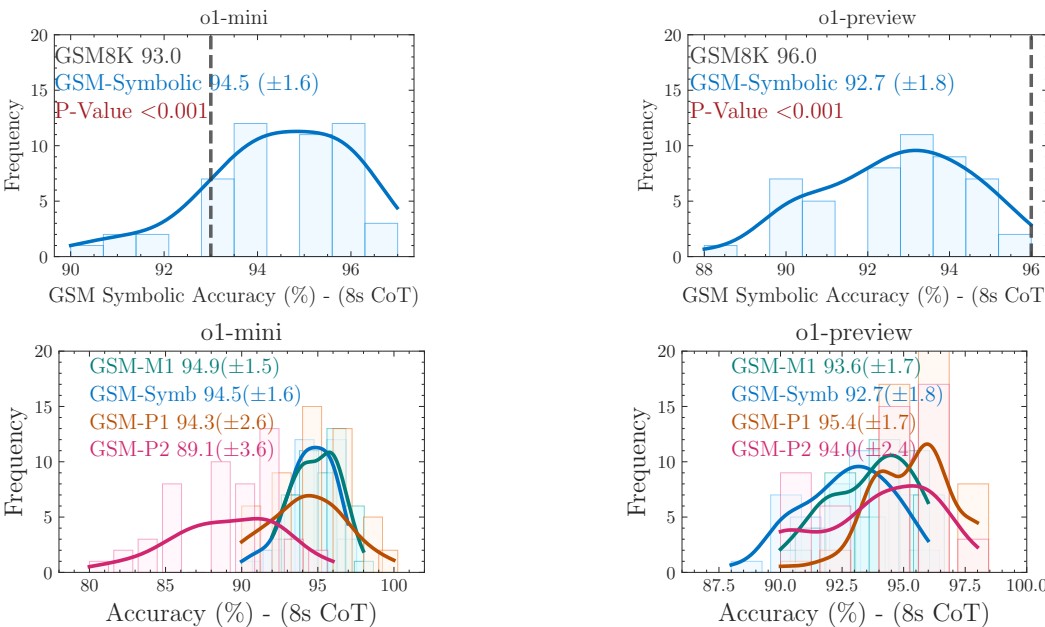

Figure 12: Results on o1-mini and o1-preview: both models mostly follow the same trend we presented in the main text. However, o1-preview shows very strong results on all levels of difficulty as all distributions are close to each other.

The o1-preview model demonstrates robust performance across all levels of difficulty, as indicated by the closeness of all distributions. However, it is important to note that both o1-preview and o1-mini experience a significant performance drop on GSM-NoOp . In Fig. 13, we illustrate that o1-preview struggles with understanding mathematical concepts, naively applying the 10% inflation discussed in the question, despite it being irrelevant since the prices pertain to this year. Additionally, in Fig. 14, we present another example highlighting this issue.

Overall, while o1-preview and o1-mini exhibit significantly stronger results compared to current open models—potentially due to improved training data and post-training procedures—they still share similar limitations with the open models.

**GSM-NoOp**

Liam wants to buy some school supplies. He buys 24 erasers that `now` cost $6.75 each, 10 notebooks that `now` cost $11.0 each, and a ream of bond paper that `now` costs $19. How much should Liam pay `now`, `assuming that due to inflation, prices were 10% cheaper last year?`

- - - - - - - - - - - - - - - - - - - - - - - - - - - - - - - - - - - - - - - - - - - - - - -

`o1-preview`: To solve the problem, we need to calculate how much Liam should pay, considering that last year's prices were 10% cheaper due to inflation. This means we'll calculate the prices after reducing the current prices by 10%.

Step 1: Calculate last year's prices by reducing the current prices by 10%.
- Eraser price last year: $6.75 × 0.90 = $6.075
- Notebook price last year: $11.00 × 0.90 = $9.90
- Bond paper price last year: $19.00 × 0.90 = $17.10

Step 2: Calculate the total cost using last year's prices.
- Total cost for erasers: 24 erasers × $6.075 = $145.80
- Total cost for notebooks: 10 notebooks × $9.90 = $99.00
- Total cost for bond paper: 1 ream × $17.10 = $17.10

Step 3: Sum up all the costs to get the total amount Liam should pay.
$145.80 + $99.00 + $17.10 = $261.90

Answer:$261.90

Figure 13: Sample response from o1-preview on an example from GSM-NoOp: the model blindly applies the inflation rate, even though the inflation amount is irrelevant as the question clearly indicates the given prices are for "now" and not last year.

---

**GSM-NoOp**

A loaf of sourdough at the cafe costs \$9. Muffins cost \$3 each. If we purchase 10 loaves of sourdough and 10 muffins, how much more do the sourdough loaves cost compared to the muffins, if we plan to donate 3 loaves of sourdough and 2 muffins from this purchase?

- - - - - - - - - - - - - - - - - - - - - - - - - - - - - - - - - - - - - - - - - - - - - - - - - - - - - - - - -

`o1-preview`:
To solve this problem, we need to compare the total cost of the sourdough loaves and muffins after considering the donated items.

Step 1: Calculate the number of items kept after donations.
- Sourdough loaves kept: $10 \text{ loaves} - 3 \text{ donated} = 7 \text{ loaves}$
- Muffins kept: $10 \text{ muffins} - 2 \text{ donated} = 8 \text{ muffins}$

Step 2: Calculate the total cost of the items kept.
- Total cost of sourdough loaves: $7 \text{ loaves} \times \$9 = \$63$
- Total cost of muffins: $8 \text{ muffins} \times \$3 = \$24$

Step 3: Find the difference in total cost.
- Difference: $\$63 - \$24 = \$39$

Answer:\$39

`o1-mini`:
Let's break down the costs step by step to determine how much more the sourdough loaves cost compared to the muffins after accounting for the donations.

1. Initial Purchase Costs
Sourdough Loaves:
Price per loaf: \$9
Quantity purchased: 10
Total Cost: 10 loaves * \$9 = \$90
Muffins:
Price per muffin: \$3
Quantity purchased: 10
Total Cost: 10 muffins * \$3 = \$30
2. Donated Items and Their Value
Donated Sourdough Loaves:

Quantity donated: 3
Value: 3 loaves * \$9 = \$27
Donated Muffins:

Quantity donated: 2
Value: 2 muffins * \$3 = \$6
3. Net Costs After Donation
To find the effective cost of what you're actually keeping (i.e., not donating):

Sourdough Loaves:
Quantity kept: 10 purchased - 3 donated = 7
Net Cost: \$90 (total) - \$27 (donated) = \$63

Muffins:
Quantity kept: 10 purchased - 2 donated = 8
Net Cost: \$30 (total) - \$6 (donated) = \$24

4. Difference in Net Costs
Sourdough Loaves: \$63
Muffins: \$24
Difference: \$63 - \$24 = \$39

Conclusion
After accounting for the donations, the sourdough loaves cost \$39 more than the muffins.

Figure 14: Sample response from o1-preview and o1-mini on an example from GSM-NoOp: while the donation amount is irrelevant to the price difference, the models subtract the amount we donate.

## A.6 ABLATION: THE IMPACT OF ARITHMETIC ACCURACY

An important question regarding the design of GSM-Symbolic is: when designing templates, "how should the numerical range of the variables be chosen?" and "how much of the performance drop can be attributed to arithmetic mistakes?".

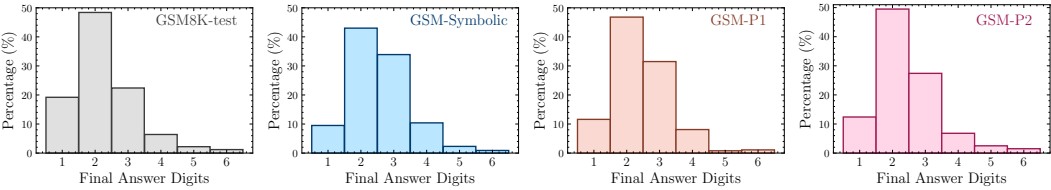

Figure 15: Statistics of the number of digits in the *final answer* for questions in GSM8K-test, GSM-Symbolic, GSM-P1, and GSM-P2. Compared to GSM8K-test, the GSM-Symbolic set shows a slight reduction in the number of questions with 1-digit or 2-digit final answers, and an increase in 3-digit answers. However, overall, the range of numbers does not increase significantly, with a significant majority of the final answers containing fewer than 5 digits.

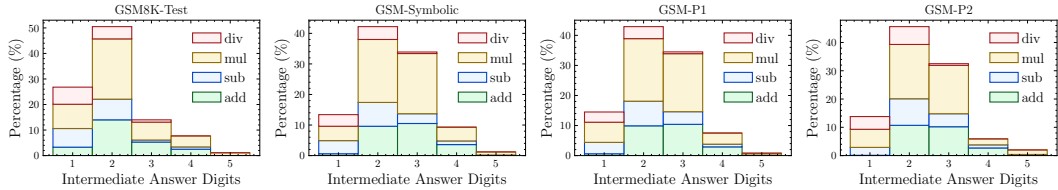

Figure 16: Statistics on the Number of Digits in *Intermediate answers*: Compared to GSM8K, GSM-Symbolic and its variants involve operations that result in 3-digit numbers. However, as shown in Fig. 17, modern LLMs are well capable of performing arithmetic within this range.

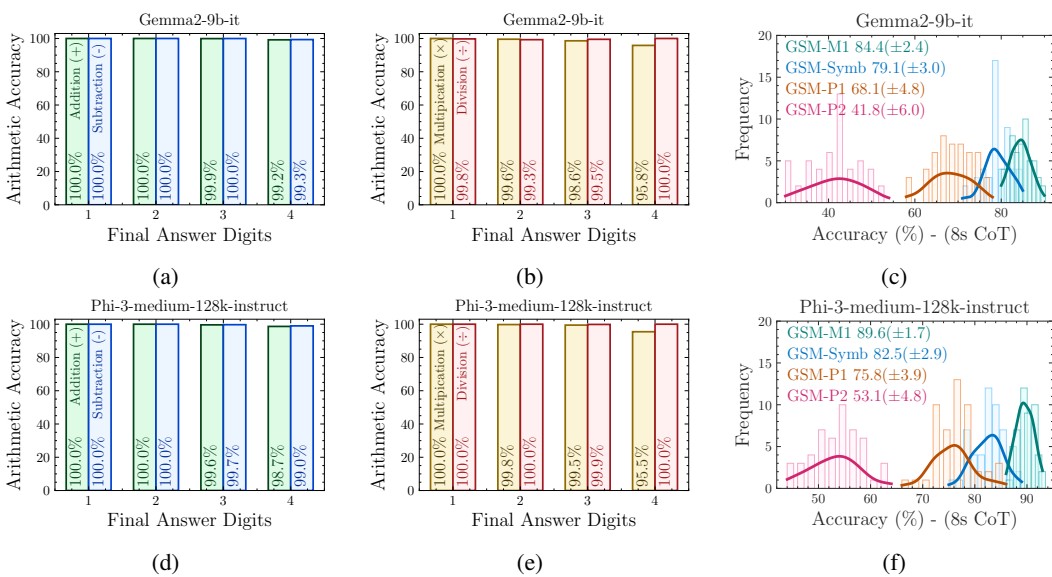

Figure 17: Arithmetic accuracy of Gemma2-9B and Phi3-Medium: **(a) and (d)** The addition accuracy of the models remains nearly perfect for calculations involving up to 4 digits. Note that, as shown in Fig. 15, the *majority of the final answers in GSM-Symbolic and its variants are below 5 digits.* **(b) and (e)** Although GSM-P1 and GSM-P2 have a similar digit-length distribution to GSM-Symbolic, as illustrated in Fig. 15, the performance drop is significant.

Overall, when creating the symbolic templates, we have chosen numerical ranges close to their original values in the GSM8K test set. The rationale behind this choice is that the arithmetic ability of the models is much less important than their logical reasoning capabilities. However, we face a challenge: we need to generate many instances per question. Given that not all values in the range satisfy the conditions and hence won't be selected, we need to slightly increase the numeric range of numbers to ensure enough instances are generated from each template.

However, we show that this adjustment does not push the numerical values into a range where models have low arithmetic accuracy. As observed in Fig. 15:

- Compared to GSM8K, GSM-Symbolic has more 3-digit final answers and fewer 1-digit and 2-digit answers. However, as shown in Fig. 17, modern LLMs such as Gemma2-9B have no difficulty with up to 3-digit addition and multiplication.
- Even though GSM-P1 and GSM-P2 have a very similar digit distribution to GSM-Symbolic and are mostly within the range that modern LLMs can perform accurate arithmetic (Fig. 17), the performance drop on these benchmarks is very significant (Fig. 17c).

Additionally, Fig. 16 provides statistics on the different operations involved in calculating *intermediate* answers across various datasets. Here, "intermediate" refers to the result of each intermediate operation throughout the solution process. For example, if a solution step involves $12 \times 6 = 72$, we categorize this as a two-digit multiplication. We observe that, overall, the distribution of the digit lengths in intermediate answers in Fig. 16, is similar to that in the final answers in Fig. 15.

Moreover, in Fig. 17, we demonstrate that modern Language Learning Models (LLMs) such as Gemma2-9B and Phi3-Medium are capable of nearly perfect addition and subtraction up to a length of 4, as well as achieving very high accuracy in multiplication and division. To this end, for each digit length of the answer, we ask the model in a zero-shot manner what the result of the operation would be. For example, for addition, we use the prompt *"What is x plus y?"* and for multiplication, we ask *"What is x times y?"*, and so on. To reduce the computational burden, we do not check arithmetic accuracy on every possible combination of numbers. However, we ensure that for every digit length and every operation, we have at least 1,000 test cases.

Finally, in Table 2 we report the arithmetic accuracy results of Gemma2-9b-it on a more realistic setup where calculations are extracted from the generated responses. To this end, we extracted parts of the generated responses where the equal sign (=) was generated along with numerical values, and then evaluated whether the left-hand side of the equation had the same numerical value as the right-hand side. As shown in the table, we observe very high arithmetic accuracy, similar to the previous results in this section.

Table 2: Ablation: Arithmetic accuracy of Gemma2-9b-it on different GSM benchmarks.

| Benchmark | 1-digit | 2-digits | 3-digits | 4-digits | 5-digits | All |
|---|---|---|---|---|---|---|
| GSM8K | 99.8 | 99.1 | 99.3 | 96.8 | 95.9 | 98.9 |
| GSM-Symbolic | 99.7 | 99.2 | 97.6 | 97.2 | 99.1 | 98.3 |
| GSM-P1 | 97.5 | 98.6 | 98.7 | 96.2 | 99.5 | 97.4 |
| GSM-P2 | 96.2 | 98.6 | 97.5 | 96.4 | 99.2 | 97.1 |

Overall, we believe it is unlikely that arithmetic difficulty can account for the significant performance drop on benchmarks such as GSM-P2, which has a very similar distribution to GSM-Symbolic and falls well within the range of fewer than 5 digits that models like Gemma2-9B can handle accurately.

