# OpenReview forum: "GSM-Symbolic: Understanding the Limitations of Mathematical Reasoning in Large Language Models"
_ICLR.cc/2025/Conference — ICLR 2025 Poster_

### Official Review · Reviewer_FK55 · 2024-10-31

**Soundness:** 2
**Presentation:** 2
**Contribution:** 2
**Rating:** 3
**Confidence:** 4

**Summary:**

The paper "GSM-SYMBOLIC: Understanding the Limitations of Mathematical Reasoning in Large Language Models" explores the mathematical reasoning capabilities of Large Language Models (LLMs) through the introduction of a new benchmark, GSM-Symbolic. This benchmark aims to address the limitations of existing evaluations like GSM8K by generating diverse question variants and adjusting complexity levels. The study finds that LLMs exhibit performance variance when faced with different instantiations of the same question, particularly when numerical values are altered. It also highlights the fragility of LLMs' reasoning capabilities, as performance deteriorates with increased question complexity and the introduction of irrelevant information. The paper suggests that current LLMs rely on probabilistic pattern-matching rather than robust logical reasoning.

**Strengths:**

- The study provides empirical evidence of the performance variance and fragility of LLMs when faced with different question instantiations and increased complexity in grade school math questions.
- The introduction of the GSM-Symbolic benchmark allows for the creation of diverse instances of the GSM8K dataset, which helps avoid data contamination and enables a more comprehensive evaluation of LLMs' reasoning capabilities.

**Weaknesses:**

- The findings largely reconfirm known limitations of LLMs, such as their reliance on probabilistic pattern-matching and sensitivity to small changes in question statements, which have been explored in depth in previous works, notably in "Are NLP Models really able to Solve Simple Math Word Problems?"
- The methods used to curate the dataset, such as altering numerical values and adding irrelevant information, are not novel and have been employed in other studies, such as "PAL: Program-aided Language Models," which introduces GSM8k-hard.
- The paper uses ambiguous terms like "true logical reasoning," "genuine mathematical reasoning," and "formal [reasoning] in the common sense term" without providing clear definitions of these terms.
- The paper’s findings on performance degradation with increasing question complexity have been similarly explored in other works as compounding errors in multi-step reasoning processes.
- The authors suggest that  performance of LLMs on GSM-Symbolic compared to their performance on GSM8k indicates significant limitations in LLMs’ ability to perform genuine mathematical reasoning. However,  performance of GPT4-o remains robust on GSM-Symbolic, as shown in figure 3, suggesting that the performance variance resulting from different numerical instantiation of the questions may be resolved with scaled training or model size and this limitations may not be inherent to all LLMs.

**Questions:**

1. Can you clearly define "true logical reasoning"? What conditions need to be satisfied for a language model to demonstrate true logical reasoning?
2. Why was the performance of o1 not investigated in certain parts of the study, such as in Figure 3?
3. Given that some models like GPT4-o show robustness across numerical changes, do the authors believe that scaled training and model size can alleviate the identified limitations? If so, how does this impact the generalizability of the findings?

---

> ### Author Response · Authors · 2024-11-16
> **Response to Reviewer FK55 (part 1)**
>
> We sincerely appreciate the reviewer's thoughtful feedback on our paper. We are eager to engage in a constructive discussion as there is still ample time remaining in the discussion period. Below, we provide a detailed response to your comments:
>
>
> ---
> > The findings largely reconfirm known limitations of LLMs, such as their reliance on probabilistic pattern-matching and sensitivity to small changes in question statements, which have been explored in depth in previous works.
>
> Thank you for highlighting this important aspect. As pointed out, several studies have examined LLM sensitivity to tokens (e.g., [1]), and we have clearly addressed this on the first page of our paper (lines 49-51) and in the related works section (Section 2).
> However, it is crucial to distinguish that our primary contribution is not merely identifying token bias or sensitivity. Rather, we offer a more nuanced and comprehensive evaluation of LLMs' mathematical reasoning abilities. To our knowledge, prior research has primarily relied on a single accuracy metric. In contrast, our work presents an analysis of “performance distribution” across diverse configurations, providing a more reliable criterion than a single metric. For example:
>
> * In Section 4.2, we investigate the impact of altering only proper names versus changes in values or numbers.
> * In Section 4.3, we assess the influence of question difficulty, demonstrating that as difficulty increases, the performance not only declines (indicating a shift in the performance distribution) but the variance also increases. To our knowledge, this variation in distribution variance has not been extensively explored before.
>
> Beyond these technical contributions, our work challenges the prevailing belief among many researchers that LLMs possess strong mathematical capabilities (e.g., see [2]). We contribute valuable evidence demonstrating instances where LLMs fall short in mathematical reasoning tasks, aspiring to catalyze an important discussion within the research community.
>
> Thank you once again for your constructive feedback. We hope our response clarifies the unique contributions of our work and encourages you to view our findings as a significant step forward in understanding and evaluating the mathematical reasoning capabilities of LLMs.
>
> References:
> [1] Jiang, Bowen, et al. "A Peek into Token Bias: Large Language Models Are Not Yet Genuine Reasoners." arXiv preprint arXiv:2406.11050 (2024).
> [2] Li, Chen, et al. "Common 7b language models already possess strong math capabilities." arXiv preprint arXiv:2403.04706 (2024).
>
> ---
>
> > The methods used to curate the dataset, such as altering numerical values and adding irrelevant information, are not novel and have been employed in other studies, such as "PAL: Program-aided Language Models," which introduces GSM8k-hard.
>
> Thank you for your insightful feedback. We understand the concern regarding the novelty of methods used to curate the dataset. Our primary aim, as highlighted in our paper, is not the mere introduction of a new dataset but rather to investigate deeper into understanding the limitations of mathematical reasoning in large language models (LLMs). While there are indeed several works that introduce new mathematical reasoning benchmarks, our focus is uniquely centered on using the GSM-Symbolic variants as a nuanced tool to study LLMs more reliably.
> The novelty of our contribution lies in our detailed analysis of the evolution of performance distribution across various setups. This aspect of our work, absent in prior studies, provides significant new insights into the capabilities and limitations of LLMs in mathematical reasoning.
>
> We hope this clarifies the confusion regarding the scope and goals of our paper.

---

> ### Author Response · Authors · 2024-11-16
> **Response to Reviewer FK55 (part 2)**
>
> > The paper uses ambiguous terms like "true logical reasoning," "genuine mathematical reasoning," and "formal [reasoning] in the common sense term" without providing clear definitions of these terms. Can you clearly define "true logical reasoning"? What conditions need to be satisfied for a language model to demonstrate true logical reasoning?
>
> We sincerely thank the reviewer for this important question. We acknowledge that our work can be improved with a more rigorous discussion on the definition and expectation of the “logical reasoning”. We will add a discussion section to our paper to clarify this. Before that, we would like to converge with the reviewers during the discussion period.
>
> **What is reasoning?**
>
> To clarify, we define "logical reasoning" as the process by which an agent employs logical steps to achieve a novel goal. The emphasis on "novelty" is crucial because it helps distinguish genuine reasoning from mere memorization of solutions or responses to previously encountered, similar problems. This distinction aligns with ideas akin to the Chinese Room Argument [3], underscoring the difference between true comprehension and pattern-matching.
> Furthermore, this definition connects to several established definitions in AI literature. For instance, the “length-generalization” refers to the ability of applying known logical steps on larger inputs. Moreover, in order to successfully applying such logical steps, an agent often requires other skills such as “decomposing” the problem into smaller sub-problems, and “composing” logical steps from these sub-problems in order to solve another problem.
>
> **How to measure reasoning? How does GSM-Symbolic measure reasoning?**
>
> Regarding measurement, it is indeed challenging to devise an objective function for assigning reasoning scores and distinguishing "memorization" from "reasoning".
> However, as a simpler criterion for the scope of our work, we propose a simple heuristic that can test if the model has followed a series of logical steps to solve a problem: if a system can solve a grade-school math problem, at very least, it should also solve the problem when proper names (e.g., names of persons, foods, objects) in the question are changed. Ideally, the performance distribution variance should be zero or minimal. Yet, this is not the case, as illustrated in Figure 4 of our paper. This indicates that naive pattern-matching might better explain the results. Similar argument holds for the scenarios where we adjust the complexity (i.e., Figure 6 in Section 4.3) as we don’t expect such a sharp drop when only one or two clauses added to a relatively simple grade-school level question. We believe these are necessary conditions, therefore even though they might not be completely characterize reasoning ability, they remain essential. Additionally, Section 4.4 of our work offers more evidence supporting this claim.
>
> While we hope this addresses reviewer's question, we acknowledge the complexities in providing a definition of reasoning that everyone agrees on at the moment, a task on which the field has yet to converge. We're eager to gather your thoughts and believe continued dialogue will significantly contribute to moving towards an agreeable definition for reasoning. Thank you for your insights and for being a part of this important conversation.
>
> ---
> > The paper’s findings on performance degradation with increasing question complexity have been similarly explored in other works as compounding errors in multi-step reasoning processes.
>
> We would like to reiterate that our work's main message regarding question complexity was not simply that performance drops, but rather the magnitude of the drop and the increased variance in performance distribution (as shown in Fig. 6). We believe our work presents a more nuanced picture of how performance distribution changes across four different levels of difficulty in Section 4.3, an analysis that was not conducted in prior work.
> Overall, the fact that a system's performance (including humans) drops with increased complexity and difficulty is expected, and we did not claim this as our contribution. Instead, we emphasize the importance of understanding both the degree of performance degradation with increased difficulty and how reliably a system operates across different difficulty levels.
> We hope that this addresses reviewers comments, and we look forward to hearing back from the reviewer.
>
> ---
> References:
>
> [3] Cole, D. (2024). The Chinese Room Argument. In E. N. Zalta & U. Nodelman (Eds.), The Stanford Encyclopedia of Philosophy (Winter 2024). Metaphysics Research Lab, Stanford University. https://plato.stanford.edu/archives/win2024/entries/chinese-room/

---

> ### Author Response · Authors · 2024-11-16
> **Response to Reviewer FK55 (part 3)**
>
> > Why was the performance of o1 not investigated in certain parts of the study, such as in Figure 3?
>
> Due to space constraints in the main text, we focused on reporting a select few models. However, you'll find the comprehensive results, including those for the o1 models, detailed in the appendix, specifically in Table 1. Additionally, Figures 6 and 8 of the main paper illustrate the results for o1-mini and o1-preview models.
>
> We also highlighted specific challenges associated with o1 models in our paper. Besides the API quota limitations, a significant hurdle is the lack of a "greedy-decoding" setting in the current OpenAI API, which introduces some randomness in generating responses.
> While o1 models do outperform other open models, we believe this is largely due to the breadth of their training data. As depicted in Figure 8, their performance notably drops in more challenging setups like GSM-NoOp, reflecting their limitations.
>
> Thank you once again for raising this point. We will incorporate an expanded discussion on the performance of the o1 models in our paper. We welcome any further questions you might have and look forward to engaging with you during the discussion period.
>
>
> > The authors suggest that performance of LLMs on GSM-Symbolic compared to their performance on GSM8k indicates significant limitations in LLMs’ ability to perform genuine mathematical reasoning. However, performance of GPT4-o remains robust on GSM-Symbolic, as shown in figure 3, suggesting that the performance variance resulting from different numerical instantiation of the questions may be resolved with scaled training or model size and this limitations may not be inherent to all LLMs.
>
> > Given that some models like GPT4-o show robustness across numerical changes, do the authors believe that scaled training and model size can alleviate the identified limitations?
>
> Thank you for raising this important question, which addresses a central focus of our work. While some argue that increasing model sizes and training data can overcome current model limitations, our analysis suggests otherwise. Though one could argue that a 10-trillion parameter LLM trained on 100 trillion tokens might perform reasoning effectively—a claim that can only be definitively refuted through empirical testing—there is currently no evidence supporting this scaling hypothesis.
>
> Our examination of model size scaling, using results from Gemma-2 2B/9B/27B in Table 1, suggests that simply increasing model size and training tokens does not overcome these reasoning limitations. Even if one accepts the scaling hypothesis, extrapolating from Table 1 results for challenging benchmarks like Symbolic-P2 and Symbolic-NoOp indicates that achieving near-perfect accuracy would require significant compute and data resources.
>
> Furthermore, there is currently no evidence that such scaled systems would generalize to more difficult or advanced mathematical questions. Regarding the GPT-4o and o1 models, since the specifics of the training data used for large closed models are unknown, it's possible that these models have been exposed to multiple variations of training examples from benchmark datasets. This exposure could potentially make them seem more robust to changes. However, this might simply shift the distinction between "pattern matching" and genuine reasoning on new problems. This is particularly evident when we increase the complexity of problems—which would still be simple for grade school students—or when we introduce confounding clauses.
>
> In conclusion, while scaling the quality and quantity of data and compute (both training and inference/test-time) can improve models' pattern-matching abilities, we believe the AI field needs to move beyond scaling and invest more in algorithmic research (e.g., different training objectives). This argument applies to frontier models such as GPT-4 as well, and notably, many prominent advocates of the scaling paradigm are now reconsidering their positions. Our results, combined with other works cited in our paper, suggest that addressing these inherent limitations requires more than scaling alone.
>
> Thank you once again to the reviewer for raising this important question. We will expand the discussion in the paper to address it further. We view this paragraph as just the starting point for answering such inquiries and encourage the community not only to pose these thought-provoking questions but also work in pushing the boundaries of understanding and improving LLM reasoning.
>
> We hope these clarifications can address the concerns raised and merit a reconsideration of the score by the reviewer.
>
> ---
>
> We would like to thank the reviewer once again for their valuable feedback, which can significantly improved our work.
> We hope you find our response compelling enough to consider a re-evaluation of our score. If there are any remaining concerns, we would really appreciate the opportunity to discuss them further during the discussion period.

---

> ### Author Response · Authors · 2024-11-28
>
> We would like to kindly follow up regarding our response to the reviewer. We truly value the reviewer's feedback and are eager to address any remaining concerns you might have during the remainder of the discussion period.

---

### Official Review · Reviewer_wG4b · 2024-11-01

**Soundness:** 3
**Presentation:** 2
**Contribution:** 2
**Rating:** 5
**Confidence:** 4

**Summary:**

This paper investigates the reasoning capabilities of LLMs using the GSM8K benchmark and introduces a new benchmark called GSM-Symbolic. The authors highlight the variability in model performance, particularly noting that LLMs struggle more when numerical values are altered compared to when only proper names are changed. They also explore the impact of question difficulty on performance, revealing that as complexity increases, both average performance and variance decline. The findings emphasize the need for more reliable evaluation methodologies and further research into the reasoning abilities of LLMs, suggesting that while these models show potential, their understanding of mathematical concepts remains limited

**Strengths:**

+ GSM-Symbolic provides a strategy to enrich the math reasoning datasets. The symbolic template can be applied for other datasets.
+ The NoOp design is innovative, it shows undiscussed disadvantage of current LLM on reasoning tasks.
+ The experiment design is adequate and valid. This paper evaluates 25 models, the results provide a comprehensive overview of the current SOTA methods.

**Weaknesses:**

- Figure 3 shows the chatgpt-o1 and GPT-4o has tiny accuracy drop. But even without GSM-Symbolic, running any LLM models for 50 times on the same dataset can cause this variance.
- For numerical variables, how did the authors choose their boundary? This is not explained in the paper
- To add clause in GSM-NoOp, we can see results drop a lot in Figure 8, but what is the standard to add this extra clause? Because adding clause can be subjective if authors already know some patterns will make the model fail.
- Part of the beginning paragraph of section 3 should belong to related work discussion.

**Questions:**

1. For numerical variables, how did the authors choose their boundary? Why x belongs to (5, 100) in Figure 1? I think this range may also influence the performance, especially when some extreme values are selected.
2. To add clause in GSM-NoOp, we can see results drop a lot in Figure 8, but what is the standard to add this extra clause? Because adding clause can be subjective if authors already know some patterns will make the model fail.

---

> ### Author Response · Authors · 2024-11-16
> **Response to Reviewer wG4b**
>
> We greatly appreciate the reviewer's thoughtful feedback and the time they've invested in evaluating our paper. We hope our responses to the comments below will effectively address any concerns and encourage a reconsideration of the score.
>
> ---
>
> > Figure 3 shows the chatgpt-o1 and GPT-4o has tiny accuracy drop. But even without GSM-Symbolic, running any LLM models for 50 times on the same dataset can cause this variance.
>
> Regarding the negligible performance drop of GPT-4o and o1 models in Figure 3, we note that limited information about these models' data and training regimes makes it challenging to provide definitive explanations. However, one possibility is that these models might have been trained on data similar to GSM-Symbolic. Indeed, as observed in more complex setups like GSM-Symbolic-P2 and GSM-Symbolic-NoOp, their performance starts to decline, suggesting a limitation on unfamiliar data structures.
>
> In addressing the variance, we clarify that, as explained in the paper (Section 3.2 and the appendix), we employ a "greedy decoding" strategy which results in consistent outputs across multiple runs on the same inputs. We specifically chose this method because, for tasks involving mathematical or logical sequences, such as GSM8K, alternative sampling strategies for decoding are not necessarily correct. For instance, with an input like "10 + 5 =", there is a singular correct solution, "15", making token sampling irrelevant in this context.
>
> We hope this response addresses your concerns, and we welcome any further discussion during the discussion period if additional clarification would be beneficial. Thank you for your constructive feedback.
>
> ---
>
> > For numerical variables, how did the authors choose their boundary? This is not explained in the paper
>
> Thank you for highlighting this important point. The numerical boundaries for variables were selected to maintain consistency in the computational order (e.g., 2-digit, 3-digit), though this wasn't always feasible for every scenario. Certain questions required specific constraints, such as ensuring the product of two variables divides a given number, thus keeping the final result an integer.
> We recognize that the increased digit length might have a slight effect (typically just one additional digit). It’s also noteworthy that even if numerical values remain constant, and only names are changed (as seen in Figure 4), notable variance can still occur. Furthermore, state-of-the-art models today comfortably manage operations involving up to 3-4 digit numbers due to memorization.
> In response to your suggestion, we will include this explanation and add an ablation study on the arithmetic accuracy of several models in the appendix. We appreciate your feedback and hope these enhancements provide clarity and justify a reconsideration of our score.
>
> ---
>
> > To add clause in GSM-NoOp, we can see results drop a lot in Figure 8, but what is the standard to add this extra clause? Because adding clause can be subjective if authors already know some patterns will make the model fail.
>
> Thank you for your insightful comment regarding the addition of clauses and the potential for subjectivity. We understand your concern and want to clarify that our approach did not involve any automated search to identify prompts that might cause all models to fail. Instead, the GSM-NoOp dataset was consistently used across all models without any tuning that might skew results in favor of or against specific models. We will add this explanation to the paper to enhance the clarity of our methodology. We hope this addresses your concern.
>
> ---
>
> > Part of the beginning paragraph of section 3 should belong to related work discussion.
>
> We agree that the paragraph discusses prior work on the GSM8K dataset and serves as background information. Our intention was to improve readability by providing context in section 3, as it transitions into discussing the GSM-Symbolic dataset. However, if you feel that this affects the flow, we are more than willing to move it to section 2 to enhance clarity.
>
> ---
>
> We appreciate your valuable feedback and believe our response has addressed your concerns, warranting a re-evaluation of the score. We welcome further discussion of any remaining questions during the discussion period. Thank you for your time and consideration.

---

> > ### Comment · Reviewer_wG4b · 2024-11-22
> > **Following question on the added clauses**
> >
> > Thanks for authors' responses, they are very informative. But still, I want to ask, how is the added clause constructed? I understand the authors did not do any automated search, but how are these clauses are designed and generated? I think this process can influence the results.

---

> > > ### Author Response · Authors · 2024-11-22
> > > **Response to Reviewer wG4b**
> > >
> > > Thank you for continuing the discussion with us.
> > >
> > > The clauses were added by an expert human based on heuristics that were inconsequential to the logic of the problem. For instance, the size of objects being purchased does not impact the number of items being purchased; the fact that the prices were different in the previous year or that someone loses or breaks items after purchase does not change the amount they paid at the time of purchase.
> > >
> > > Hope this clarifies the reviewer's questions.

---

> ### Author Response · Authors · 2024-11-22
> **Update for Reviewer wG4b**
>
> We would like to notify the reviewer that we have revised the paper to include comprehensive results in Appendix A.6, focusing on how the arithmetic accuracy of LLMs affects our findings.
>
> Our results indicate that the numerical range for GSM-Symbolic and its variants (Figures 15 and 16) falls well within the scope where LLMs can accurately perform arithmetic (Figure 17). Consequently, we demonstrate that, despite GSM-P1/P2 having nearly identical digit-length distributions in both their intermediate and final answers compared to GSM-Symbolic, the notable decline on these benchmarks (Figure 17 (c) & (f)) is unlikely due to arithmetic errors.
>
> We appreciate the reviewer's question on this matter, and we believe our updated findings have strengthened our paper. We hope this new analysis has addressed the reviewer's concerns about the impact of impact of numerical boundaries and deserves a re-evaluation of the score. Lastly, we welcome any further feedback from the reviewer throughout the remaining discussion period.

---

> > ### Author Response · Authors · 2024-12-02
> >
> > As the discussion period nears its end, we wanted to kindly follow up to see if you've had a chance to review our response and new revision. We would appreciate knowing if it addressed your concerns or there are any remaining concerns that we can address.
> >
> > Best Regards,
> >
> > Authors

---

### Official Review · Reviewer_7pZc · 2024-11-04

**Soundness:** 3
**Presentation:** 4
**Contribution:** 3
**Rating:** 8
**Confidence:** 4

**Summary:**

The paper "GSM-Symbolic: Understanding the Limitations of Mathematical Reasoning in Large Language Models" presents a thorough investigation into the mathematical reasoning capabilities of large language models (LLMs) using a newly developed benchmark, GSM-Symbolic. This benchmark, derived from the GSM8K dataset, includes modifications to test the robustness of LLMs against changes in numerical values, proper names, and question complexity.

**Key Contributions:**

1. **Introduction of GSM-Symbolic**: The paper introduces GSM-Symbolic, a benchmark that generates diverse and controllable mathematical question variants based on symbolic templates. This allows for more precise evaluations of LLMs by modifying elements like numbers and text within the questions to assess their impact on model performance.

2. **Empirical Analysis of LLM Performance**: Through extensive testing with GSM-Symbolic, the authors demonstrate that LLMs exhibit significant performance variability and sensitivity to minor changes in question structure. Their experiments reveal that even small perturbations, such as altering numerical values or adding irrelevant clauses, can significantly degrade the performance of state-of-the-art models.

3. **Exploration of Reasoning Fragility in LLMs**: The study details how the introduction of irrelevant information (via the GSM-NoOp dataset) into mathematical problems significantly impacts model accuracy, highlighting a critical flaw in the current models' ability to discern relevant from irrelevant data.

Overall, the paper provides a deeper understanding of the limitations inherent in current LLMs regarding their mathematical reasoning capabilities and offers a comprehensive framework for future research to explore and mitigate these weaknesses.

**Strengths:**

**Strengths:**

1. **Comprehensive Experimental Design**: The authors explore various perturbations in mathematical questions such as changing names or numbers, adjusting the number of clauses, and introducing irrelevant information. These detailed experiments provide robust evidence of the fragility exhibited by prominent language models under slight modifications, thereby contributing significantly to our understanding of their limitations.

2. **Development of the GSM-Symbolic Dataset**: The introduction of the GSM-Symbolic dataset and its variants is a notable contribution. By coding 100 problems from the GSM8K dataset and creating corresponding modifications, the authors have developed a valuable resource that can facilitate future research into model robustness.

3. **Relevance and Depth of Analysis**: The topic is highly relevant to current discussions in the field, and the paper delivers extensive empirical data concerning critical questions about the robustness of current language models when subject to various perturbations.

**Weaknesses:**

**Weaknesses:**

1. **Oversight of Computational Complexity**: The paper does not sufficiently address the computational challenges posed by the range of parameter values (5 to 100) used in their experiments. This range suggests that computing expressions like x + y + z, particularly with all parameters as two-digit numbers, could introduce significant computational errors unrelated to the models' reasoning capabilities. The potential impact of these computational difficulties on the results, particularly evident in Figure 4, suggests that the observed distribution shifts in Figure 2 might be attributed more to computational challenges than to data contamination.

2. **Ambiguity in Defining Reasoning**: The paper's definition of reasoning needs clarification. Asserting that models capable of only three-step inferences lack reasoning capabilities seems overly restrictive. Since the GSM8K focuses on the NLP aspect of reasoning rather than the length of reasoning required, it naturally tailors models to excel within these confines. Thus, introducing additional clauses or irrelevant details, as done in the experiments, effectively renders the questions out-of-distribution (OOD). This shift likely degrades performance due to the models not being trained to handle such modifications, which doesn't necessarily indicate a lack of reasoning ability but rather a limitation in the training paradigm.

**Questions:**

**Q1: Computational Complexity Impact**
Could the authors provide a detailed analysis of how computational complexity affects the accuracy of the models tested? Specifically, it would be informative to understand the proportion of errors that can be attributed directly to the failure of numerical computations versus those caused by reasoning errors. This differentiation could help clarify the extent to which computational demands influence the performance of large language models on the GSM-Symbolic dataset.

**Q2: Impact of Fine-Tuning on Model Robustness**
Would the authors consider conducting experiments where some models are fine-tuned on the GSM-Symbolic dataset and its variants to assess whether this training approach enhances their robustness to the types of perturbations introduced in your study? This could provide valuable insights into whether the observed limitations are inherent to the model architectures or can be ameliorated through targeted training, potentially challenging the paper's conclusions about the fundamental nature of these limitations.

---

> ### Author Response · Authors · 2024-11-16
> **Response to Reviewer 7pZc (part 1)**
>
> Thank you for your insightful feedback. We've carefully addressed your comments below and hope that we can discuss any remaining concerns during the discussion period to support a positive re-evaluation of the score.
>
> ---
>
> > Oversight of Computational Complexity: The paper does not sufficiently address the computational challenges posed by the range of parameter values (5 to 100) used in their experiments. This range suggests that computing expressions like x + y + z, particularly with all parameters as two-digit numbers, could introduce significant computational errors unrelated to the models' reasoning capabilities. The potential impact of these computational difficulties on the results, particularly evident in Figure 4, suggests that the observed distribution shifts in Figure 2 might be attributed more to computational challenges than to data contamination.
>
> > Q1: Computational Complexity Impact Could the authors provide a detailed analysis of how computational complexity affects the accuracy of the models tested? Specifically, it would be informative to understand the proportion of errors that can be attributed directly to the failure of numerical computations versus those caused by reasoning errors. This differentiation could help clarify the extent to which computational demands influence the performance of large language models on the GSM-Symbolic dataset.
>
>
> Thank you for raising this important point. We selected numerical boundaries for the variables to ensure computational consistency, aiming to keep arithmetic operations at similar digit levels. While this wasn’t always possible for every question (e.g., due to the need for specific constraints to make the product of two variables an integer), we made slight adjustments to the numerical range to generate a sufficient number of questions.
> We acknowledge that this approach might increase digit length slightly, often by just one digit. It's worth mentioning that even when numerical values stay the same and only the names change (as shown in Figure 4), significant variances can still occur. Additionally, modern state-of-the-art models handle operations involving 3-4 digit numbers quite effectively because of their memorization capabilities.
> Based on your suggestion, we'll incorporate this explanation and add an ablation study on the arithmetic accuracy of various models in the appendix. We truly appreciate your feedback and hope these enhancements clarify our approach and encourage a reconsideration of our score.
>
> ---
>
> > Since the GSM8K focuses on the NLP aspect of reasoning rather than the length of reasoning required, it naturally tailors models to excel within these confines. Thus, introducing additional clauses or irrelevant details, as done in the experiments, effectively renders the questions out-of-distribution (OOD). This shift likely degrades performance due to the models not being trained to handle such modifications, which doesn't necessarily indicate a lack of reasoning ability but rather a limitation in the training paradigm.
>
> Thank you for your thoughtful feedback. We understand your concerns regarding the out-of-distribution (OOD) setup. However, we'd like to clarify that the additional clauses introduced to modify complexity in GSM-P1/P2 (as seen in Figure 5) don't necessarily render the questions as OOD, as these clauses closely resemble those in GSM8K.
>
> We agree that GSM-NoOp scenarios can be viewed as OOD, as existing models might not have been trained on such datasets. However, we note that even in this scenario, our results in Section 4.4 show that the models fail to answer questions correctly, even when in-context shots include solutions to the very same question.
>
> Our main focus in sections 4.3 and 4.4 is not solely on the fact that performance drops with added complexities—this was anticipated. Instead, we want to emphasize the significant magnitude of the drop with only slight modifications. If we look at the examples in Figure 5 for GSM-P2, the addition of steps doesn't fully account for the nearly 40% performance drop in Gemma2-9B and Phi-3.5.
> An intelligent system capable of reasoning is expected to solve similar problems, provided it has the requisite knowledge. For instance, after adequate training with grade-school math data, a minor addition of one or two clauses shouldn't significantly impact performance (unlike our observation in Section 4). Otherwise, training a model on all potential scenarios across every domain becomes computationally impractical—a situation we find with current models.
>
> We hope this addresses your concern, and we look forward to discussing this crucial issue further during the discussion period. Reasoning and its relation with in- and out-of-distribution generalization is an important question for the research community and we look forward to hearing more works along this line. Thanks for pointing this out.

---

> ### Author Response · Authors · 2024-11-16
> **Response to Reviewer 7pZc (part 2)**
>
> > Ambiguity in Defining Reasoning: The paper's definition of reasoning needs clarification. Asserting that models capable of only three-step inferences lack reasoning capabilities seems overly restrictive.
>
> Thank you for this important question about defining logical reasoning. We plan to add a dedicated section addressing this in our paper. Before making these revisions, we'd like to align our understanding through this discussion.
>
> **What is reasoning?**
>
> To clarify, we define "logical reasoning" as the process by which an agent employs logical steps to achieve a novel goal. The emphasis on "novelty" is crucial because it helps distinguish genuine reasoning from mere memorization of solutions or responses to previously encountered, similar problems. This distinction aligns with ideas akin to the Chinese Room Argument [1], underscoring the difference between true comprehension and pattern-matching.
>
> Furthermore, this definition connects to several established definitions in AI literature. For instance, the “length-generalization” refers to the ability of applying known logical steps on larger inputs. Moreover, in order to successfully applying such logical steps, an agent often requires other skills such as “decomposing” the problem into smaller sub-problems, and “composing” logical steps from these sub-problems in order to solve another problem.
>
> **How to measure reasoning? How does GSM-Symbolic measure reasoning?**
>
> Regarding measurement, it is indeed challenging to devise an objective function for assigning reasoning scores and distinguishing "memorization" from "reasoning".
> However, as a simpler criterion for the scope of our work, we propose a simple heuristic that can test if the model has followed a series of logical steps to solve a problem: if a system can solve a grade-school math problem, it should also solve the problem when proper names (e.g., names of persons, foods, objects) in the question are changed. Ideally, the performance distribution variance should be zero or minimal. Yet, this is not the case, as illustrated in Figure 4 of our paper. This indicates that naive pattern-matching might better explain the results. Similar argument holds for the scenarios where we adjust the complexity (i.e., Figure 6 in Section 4.3) as we don’t expect such a sharp drop when only one or two clauses added to a relatively simple grade-school level question. Additionally, Section 4.4 of our work offers more evidence supporting this claim.
>
> Overall, we acknowledge that defining reasoning rigorously, remains an open challenge in the field. We welcome your perspective and believe this dialogue will help develop a more robust definition. Thank you for engaging in this discussion.
>
>
> [1] Cole, D. (2024). The Chinese Room Argument. In E. N. Zalta & U. Nodelman (Eds.), The Stanford Encyclopedia of Philosophy (Winter 2024). Metaphysics Research Lab, Stanford University. https://plato.stanford.edu/archives/win2024/entries/chinese-room/
>
> ---
>
> > Impact of Fine-Tuning on Model Robustness Would the authors consider conducting experiments where some models are fine-tuned on the GSM-Symbolic dataset and its variants to assess whether this training approach enhances their robustness to the types of perturbations introduced in your study? This could provide valuable insights into whether the observed limitations are inherent to the model architectures or can be ameliorated through targeted training, potentially challenging the paper's conclusions about the fundamental nature of these limitations.
>
> Thank you for raising this important question. Indeed, we have considered this important question as well and briefly explored this in appendix A.4 and we show that while finetuning on 10,000 examples from GSM-P1 can slightly improve the performance on GSM-P1 examples, it does not improve the performance on the more challenging GSM-P2.
>
> Overall, we agree that this question deserves much further investigation. It could be that given the limited amount of labeled templates we used for training, and the conclusion could change for larger scale. However, we believe given that the current state of the art models have incorporated vast amounts of high-quality annotated mathematical questions and answers, it is unlikely that the reasoning limitations of the models are due to lack of knowledge or lack of training data. Instead, our results, along with many other results across the literature, suggest the limitations need to be resolved at a more fundamental level such as training objectives beyond supervised training and finetuning.
>
> We would be glad to incorporate any specific experiments suggested by the reviewer into our next revision of the paper.
>
> ---
>
> Thank you once again for your insightful feedback, which has significantly improved our paper. We hope that our response has effectively addressed your concerns and hope the proposed changes lead to a reevaluation of the score. We truly appreciate your time and thoughtful consideration.

---

> ### Author Response · Authors · 2024-11-22
> **Response to Reviewer 7pZc (part 3) - additional results**
>
> We would like to inform the reviewer that we have updated the paper to include detailed results in Appendix A.6 regarding the impact of arithmetic accuracy of LLMs on our findings.
>
> Our results demonstrate that the numerical range in GSM-Symbolic and its variants (Figures 15 and 16) is well within the range in which LLMs can perform accurate arithmetic (Figure 17). Consequently, we show that even though GSM-P1/P2 have very similar digit-length distributions in their intermediate and final answers compared to the distributions of GSM-Symbolic, the significant drop observed on these benchmarks is unlikely to be attributed to arithmetic mistakes.
>
> We thank the reviewer again for raising this question, and we believe our new results have improved our paper. We hope this new analysis has addressed the reviewer's question regarding the impact of computational complexity and merits a re-consideration of the score. Finally, we welcome any additional feedback from the reviewer during the remainder of the discussion period.

---

> > ### Comment · Reviewer_7pZc · 2024-11-27
> >
> > **On the Definition of Reasoning**
> >
> > While I appreciate your attempt to define "logical reasoning," I find the provided definition overly broad and lacking in precision. A more rigorous approach could dissect reasoning into specific capabilities such as dictionary lookup, dependency understanding, and Chain of Thought processing (summarizing relationships in a topological order and addressing them step by step).
> >
> > Consider this hypothetical scenario:
> > Imagine a machine that has memorized 1,000 problem templates. When presented with a new problem, it searches for a matching template, differing only in numerical values and parameter names. If a match is found, the machine substitutes values and computes the answer without truly "understanding" the problem's structure.
> >
> > Can we assert that such a machine possesses reasoning abilities? It appears not. It merely replicates the solution structure without genuine comprehension. A truly reasoning machine would not simply follow memorized templates but would adapt its approach to variations in problem structure, demonstrating flexibility and understanding.
> >
> > Please consider refining your definition of reasoning to better encapsulate these nuances and further explore why language models falter, treating this as a direction for future research.
> >
> > **Computational Accuracy in Applied Contexts**
> >
> > The results in Appendix A.6 do not fully address my initial query. My concern centers on the accuracy of computations within the complete solutions generated by language models on the GSM-Symbolic dataset, rather than merely testing zero-shot arithmetic proficiency.
> >
> > 1. The complexity of computations in realistic problem-solving scenarios, such as combining three-digit operations in forms like A + B + C or (A + B) * C, significantly exceeds that of simpler A + B calculations. This complexity likely affects computational accuracy.
> > 2. There is a distinction between a model learning a skill and its ability to apply this skill effectively within complex problem-solving contexts. Demonstrating that a model can perform simple arithmetic does not necessarily translate to maintaining accuracy in the multifaceted computations required in real mathematical problems.
> >
> > An automatic process to examine and verify the accuracy of each computational step within model-generated solutions could be invaluable. If these are truly simple calculations, as demonstrated in Appendix A.6, designing such a verification program should be straightforward.
> >
> > **Conclusion**
> >
> > Despite its shortcomings, this paper provides a valuable dataset and uncovers significant phenomena warranting further exploration. While I plan to maintain the current score due to the merits already demonstrated, I encourage a more detailed exploration of the types of errors models frequently make and refinement of your experimental methods as suggested. Should you comprehensively address at least one of the two concerns raised above, I will consider raising the score by one point to reflect the improved rigor and comprehensiveness of your analysis.

---

> > > ### Author Response · Authors · 2024-11-28
> > >
> > > We thank the reviewer for their valuable feedback and hope to address their remaining concerns below:
> > >
> > > ---
> > > **On the Definition of Reasoning**.
> > > > Regarding the definition of reasoning, we agree with the reviewer. However, we note that the formal definition of reasoning that encompasses all its nuances, to the best of our knowledge, remains an open problem in the literature.
> > >
> > > We believe an ideal definition of reasoning should apply to both humans and machines. However, the functions mentioned, such as dictionary lookup, dependency understanding, and CoT, are functions that we may think a reasoner should perform. But to the best of our knowledge, there is no strong evidence that these are necessary conditions for a reasoner. While we agree with the reviewer that pinning the definition of reasoning to these functions would be more rigorous, intuitively, these functions (e.g., lookup, CoT) resemble implementation details rather than criteria to measure the correctness of a program.
> > >
> > >
> > > > Can we assert that such a machine possesses reasoning abilities? It appears not... A truly reasoning machine would not simply follow memorized templates but would adapt its approach to variations in problem structure, demonstrating flexibility and understanding.
> > >
> > > We believe that a key component of reasoning involves solving problems that exhibit combinatorial complexity, resulting in numerous potential paths to a solution. This implies (a) it's unlikely the reasoner has encountered that specific question before, and (b) the solution typically requires a sequence of decisions to break down the complex problem into a series of smaller problems that might be solved through memorization. However, it is the unique combination of these smaller solutions that constitutes reasoning.
> > > We understand that, given our definition, it is not always possible to distinguish between reasoning and memorization without further assumptions. However, there are two important points we would like to make to explain our perspective:
> > >
> > > * First, this problem may also arise when measuring the reasoning capabilities of any system, including humans. If we pose a riddle to a human without further assumptions or information, we cannot determine whether the human already knew the riddle or independently came up with the answer. This issue relates to the chinese room argument we mentioned in our initial response. This is why we emphasized the term "novelty" in our definition and related it to the goal of "variations in problem structure" in the reviewer's comment.
> > > * Second, it is possible that, without further assumptions, determining whether a system is reasoning or not is undecidable, similar to many important problems, such as the halting problem for Turing machines. We recognize that the undecidability of this problem may not be ideal, and perhaps this problem will eventually turn out to be decidable. However, as researchers, we believe we should be open to all possible outcomes, which does not necessarily imply that our definition lacks rigor.
> > >
> > > Overall, this is a very important question, and we understand that the reviewer may not be satisfied with our definition or may disagree with our perspective. However, we believe our paper can contribute to an open yet critical discussion on reasoning.

---

> > > ### Author Response · Authors · 2024-11-28
> > >
> > > **Computational Accuracy in Applied Contexts**.
> > >
> > >
> > > Regarding computational accuracy, we have a new result for the setup suggested by the reviewer. While we cannot submit a new revision anymore, we provide the details and results here.
> > >
> > > First, we calculated the arithmetic accuracy of Gemma2-9b-it based on their generated responses to several benchmarks. As suggested by the reviewer, this is a more realistic setup where calculations are extracted from the generated responses. To this end, we extracted parts of the generated responses where the equal sign (=) was generated along with numerical values, and then evaluated whether the left-hand side of the equation had the same numerical value as the right-hand side.
> > >
> > > As shown in the table below, we observe very high arithmetic accuracy, similar to our reported results in Appendix A.6. Moreover, we observe that the overall accuracy of the model remains very high on GSM-P1 and GSM-P2, despite observing a significant performance drop. In other words, GSM problems involve reasoning (i.e., decomposing novel complex problems into a sequence of simpler ones) and arithmetic (which current LLMs can accurately perform at the grade-school level). Our results in the paper and here show that reasoning is indeed different than arithmetic.
> > >
> > >
> > > | Benchmark | 1-digit | 2-digits | 3-digits | 4-digits | 5-digits | All |
> > > |---|---|---|---|---|---|---|
> > > | GSM8K | 99.8 | 99.1 | 99.3 | 96.8 | 95.9 | 98.9 |
> > > | GSM-Symbolic | 99.7 | 99.2 | 97.6 | 97.2 | 99.1 | 98.3 |
> > > | GSM-P1 | 97.5 | 98.6 | 98.7 | 96.2 | 99.5 | 97.4 |
> > > | GSM-P2 | 96.2 | 98.6 | 97.5 | 96.4 | 99.2 | 97.1 |
> > >
> > >
> > > Overall, our current results demonstrate that the significant drop in performance in more difficult scenarios is unlikely to be mainly due to arithmetic errors; however, this topic warrants further investigation. We will add these results, along with additional models, to our next revision. As noted by the reviewer, our dataset, which will be released after decisions, can be a valuable tool for researchers working at the intersection of reasoning and language models.
> > >
> > > We hope this has addressed the reviewers concern and we welcome any further feedback by the reviewer.

---

> ### Comment · Reviewer_7pZc · 2024-11-28
>
> Thank you for the detailed experiments and for considering the experiments I suggested. I appreciate the effort and responsiveness demonstrated in your revisions. Based on these updates, I'm pleased to raise my score to 8.

---

### Official Review · Reviewer_Yt9o · 2024-11-04

**Soundness:** 3
**Presentation:** 2
**Contribution:** 4
**Rating:** 8
**Confidence:** 4

**Summary:**

This paper creates a novel dataset (GSM-Symbolic) with several different variants to test whether LLMs are capable of generalized reasoning. GSM-Symbolic has several variants which change the numbers involved in the problem and/or removes or adds clauses that change the difficulty level of the problem but not in a way that is too different from the original problem. They benchmark across a number of models and show that many LLMs show significant drops when changing the numbers only slightly or adding irrelevant information.

I have some questions that I raise in the sections below. In particular, I think some of the claims are not fully supported by the provided evidence and that it's not clear that you can make strong, broad claims about LLMs as opposed to claims about the particular models tested. Nevertheless, I believe this paper to be both high quality and important work and think that it should be accepted to the conference.

Nits:

l84, citation format error
L759: spacing error

**Strengths:**

Paper tests a very good question. The state of LLMs is very strange right now. They can clearly solve very tricky problems. At the same time, they often fail in very elementary ways as well. This paper is an exceptional analysis of this, especially testing across several different variants of GSM-Symbolic. In general, I think this paper is a fantastic contribution to the field and as a result vote accept.

**Weaknesses:**

Some weaknesses are referenced in the questions section. One additional question: would the authors be able to include statistical significance results in the Appendix or main results? I think this would significantly improve the paper.

Secondly, while the GSM-Noop experiments are very interesting, I think there is a large difference in the claim (also made by prior work) that LLMs are bad at handling irrelenvant context and them not performing reasoning.

Related Work: I think Srivastava+ 2024 should be cited and compared to, as it is a very similar method on the MATH dataset with similar-ish findings.

@misc{srivastava2024functionalbenchmarksrobustevaluation,
      title={Functional Benchmarks for Robust Evaluation of Reasoning Performance, and the Reasoning Gap},
      author={Saurabh Srivastava and Annarose M B and Anto P V au2 and Shashank Menon and Ajay Sukumar and Adwaith Samod T and Alan Philipose and Stevin Prince and Sooraj Thomas},
      year={2024},
      eprint={2402.19450},
      archivePrefix={arXiv},
      primaryClass={cs.AI},
      url={https://arxiv.org/abs/2402.19450},
}

@misc{shi2023largelanguagemodelseasily,
      title={Large Language Models Can Be Easily Distracted by Irrelevant Context},
      author={Freda Shi and Xinyun Chen and Kanishka Misra and Nathan Scales and David Dohan and Ed Chi and Nathanael Schärli and Denny Zhou},
      year={2023},
      eprint={2302.00093},
      archivePrefix={arXiv},
      primaryClass={cs.CL},
      url={https://arxiv.org/abs/2302.00093},
}

**Questions:**

1. The abstract claims that “Specifically, the performance of all models declines when only the numerical values in the question are altered in the GSM-Symbolic benchmark.” However, if I’m reading this right, this is not what you report in Table 1 in A.2? For example, o1-mini and both Llama3 8B models seem to get a higher score on Symbolic compared to GSM8k (100).

2. One potential concern I have with the paper is that. Especially given (1), how do you differentiate between results that apply to LLMs more broadly compared  claims that apply broadly to specific LLMs? In particular, several models in your evaluations are well known to have interesting results regarding overfitting on GSM8k as per works in your current related works section.

---

> ### Author Response · Authors · 2024-11-22
> **Response to Reviewer Yt9o**
>
> We sincerely appreciate the reviewer's thoughtful feedback on our paper. We're delighted that you recognize the importance and high quality of our work, as well as the exceptional analysis it offers on the current state of LLMs. Below, we've addressed your comments:
>
> ---
> > Would the authors be able to include statistical significance results in the Appendix or main results? I think this would significantly improve the paper.
>
> We agree on the importance of providing statistical significance. We have updated our Appendix A.3 (Figure 10) with statistical significance results based on one-sample t-test. However, we note that this is a complicated topic that deserves much further study and we will continue our investigation. Thank you your suggestion.
>
> ---
>
> > Secondly, while the GSM-Noop experiments are very interesting, I think there is a large difference in the claim (also made by prior work) that LLMs are bad at handling irrelenvant context and them not performing reasoning.
>
> Indeed, it is difficult to disentangle the impact of irrelevant context versus limitations in reasoning.
> However, we would like to highlight the non-negligible variance reported in section 4.2 when only changing the names, in addition to significant performance drops by adding a one or two relevant clauses in section 4.3 (Figure 6). We believe the results suggest that the limitations even when there is no irrelevant clause added to the questions.
>
> ---
>
> > Related Work: I think Srivastava+ 2024 should be cited and compared to, as it is a very similar method on the MATH dataset with similar-ish findings.
>
> Thank you for your suggestions. We agree with the reviewer and we will update the the text to address these comments.
>
> ---
>
> > The abstract claims that “Specifically, the performance of all models declines when only the numerical values in the question are altered in the GSM-Symbolic benchmark.” However, if I’m reading this right, this is not what you report in Table 1 in A.2? For example, o1-mini and both Llama3 8B models seem to get a higher score on Symbolic compared to GSM8k (100).
>
> Thank you for pointing this out. We will update the abstract to be consistent with our results in Table 1.
>
> ---
>
> > One potential concern I have with the paper is that. Especially given (1), how do you differentiate between results that apply to LLMs more broadly compared claims that apply broadly to specific LLMs? In particular, several models in your evaluations are well known to have interesting results regarding overfitting on GSM8k as per works in your current related works section.
>
> Thank you for bringing up this crucial point. We agree that our current results primarily focus on benchmarks derived from GSM8K, and there are valid concerns about potential overfitting with some models on this dataset. Additionally, the lack of transparency in the training data of state-of-the-art models makes interpreting the results even more challenging. However, we've carefully designed our experiments to mitigate these concerns. For instance, we report a performance distribution rather than just a single number, providing a more comprehensive view of model performance, as illustrated in Figures 3 and 10. Additionally, the GSM-P1/P2 and GSM-NoOp variants include additional clauses not present in GSM8K, which helps test the robustness and adaptability of the models more thoroughly. Our findings demonstrate that models exhibit significant vulnerabilities when faced with data that is different than what they have encountered during training (assuming exposure to the GSM8K dataset during the training).
>
> ---
>
> We sincerely appreciate reviewer’s thoughtful feedback and hope that we have addressed their concerns in our response. We would be delighted to continue the dialogue if there are any further comments or questions.

---

> > ### Comment · Reviewer_Yt9o · 2024-11-26
> > **Thank you for your response.**
> >
> > I have no further questions at this time.

---

### Public Comment · ~Yifan_Zhang16 · 2024-11-13
**Public Comment on Similarity with TemplateGSM Dataset**

Thank you for this detailed and thought-provoking submission on GSM-Symbolic and its potential to improve the evaluation of mathematical reasoning in large language models. While the proposed extensions and findings are compelling, I would like to raise a concern regarding potential overlap with an existing dataset, **TemplateGSM** (https://huggingface.co/datasets/math-ai/TemplateGSM), hosted on Hugging Face and documented in their dataset repository (Published at January 2024).

**Specific Concerns:**

1. **Similarity in Dataset Generation Approach**:
   Both GSM-Symbolic and TemplateGSM seem to employ template-based question generation with systematic variations in numerical values and structures. TemplateGSM explicitly leverages templates to create a diverse set of mathematical problems derived from GSM8K. Could you clarify how GSM-Symbolic's symbolic templates differ in methodology and intent from TemplateGSM's approach Template-based Data Generation (https://templatemath.github.io/TemplateMath_Part_I.pdf)?

2. **Overlap with GSM8K and TemplateGSM**:
   GSM-Symbolic claims to extend GSM8K with symbolic reasoning templates. However, TemplateGSM similarly builds upon GSM8K, introducing templates with symbolic components to test reasoning. Could you elaborate on how GSM-Symbolic ensures originality and distinguishes itself from TemplateGSM?

3. **Attribution and Citation**:
   If GSM-Symbolic indirectly or directly draws inspiration from TemplateGSM, it would be helpful for the community to understand the relationship between the two datasets. Were any of the templates or ideas in GSM-Symbolic informed by TemplateGSM's work? If so, proper acknowledgment and citation would be warranted under academic and ethical standards.

I appreciate the effort and insight behind GSM-Symbolic, but addressing these questions will help clarify its novelty and ensure proper recognition of prior work. Thank you in advance for providing transparency on these matters.

---

> ### Author Response · Authors · 2024-11-14
> **Response to Public Comment on Similarity with TemplateGSM Dataset**
>
> Thank you for bringing your work to our attention. We appreciate the opportunity to clarify the distinctions and originality of our work on GSM-Symbolic relative to TemplateGSM. Below, we address your points:
>
> **Originality:**
>
> We would like to clearly state that our work was developed independently and was not informed by TemplateGSM. During our research, TemplateGSM was not readily accessible or easy to discover, as it appears to have been released without an accompanying research paper and primarily as data files on GitHub and Hugging Face. Additionally, to the best of our understanding, neither [1] nor [2] detailed the methodology for template generation, nor were the original templates themselves published. As such, it is challenging for researchers to derive inspiration or build upon your work from the information provided.
> We believe that given the lack of disclosure regarding the creation of templates in TemplateGSM [1,2], the assertion that our work drew inspiration from it is unfounded. Finally, we would like to note that the broader concept of "generating synthetic data" is a well-established idea that predates both GSM-Symbolic and TemplateGSM.
>
>
> However, 15 hours ago, following their initial comment on OpenReview, the authors updated their GitHub repository ([commit: 5589884](https://github.com/iiis-ai/TemplateMath/commit/5589884cbc62094fc084a1e2ebdb4d2ea61587b2)) to include a PDF report with additional details of their work. Curiously, the report’s title page indicates a date of February 2024, which could misleadingly suggest it was available since then. Fortunately, GitHub’s version history clearly shows it was uploaded only a few hours ago.
>
> Additionally, they modified the title of their project ([commit: 91ad0de](https://github.com/iiis-ai/TemplateMath/commit/91ad0de939bd5388176695476770d0ff4f827ce7)) by changing “syntactic” to “template-based” and adding “evaluation” to both the title and the focus of their work. The authors also revised their comment on OpenReview (visible through the [comment revision history](https://openreview.net/revisions?id=Bas5b70G6B)) to include a link to this newly added PDF, creating the impression that it has been available since February, without clarifying it was uploaded recently. They similarly modified the associated HuggingFace page.
>
> Given that this report was only uploaded *15 hours ago* and was absent elsewhere online before that, it is evident that our work could not have been influenced by it. We also encourage the comment author to engage with greater transparency and professionalism in their communications.
>
>
> **Distinct Differences in Data and Goals:**
>
> Our research offers several unique benchmarks beyond the GSM-Symbolic dataset. Unlike TemplateGSM that uses the GSM8K-*train* set, our approach is focused on evaluating models rather than training them, thereby working specifically with the GSM8K-*test* set. Beyond GSM-Symbolic, we have introduced additional datasets, including:
>
> * GSM-names: Modifies proper names while retaining original values from the GSM8K-test set.
> * GSM-vars: Alters numerical values while keeping proper names consistent with the GSM8K-test set.
> * GSM-M1: Removes one clause from GSM-Symbolic.
> * GSM-P1: Adds one clause to GSM-Symbolic.
> * GSM-P2: Adds two clauses to GSM-Symbolic.
> * GSM-NoOp: Incorporates seemingly relevant but inconsequential clauses into the questions.
>
>    Each of these datasets is built on unique templates, none of which overlap with those in TemplateGSM.
> Moreover, our primary focus is not solely on data creation but on analyzing and understanding how large language models perform mathematical reasoning. If there are any specific examples of overlap that you could share from [1,2], we would appreciate reviewing them. As mentioned before, while the idea of syntactic/symbolic generation is well explored in many prior works, our focus here was to use symbolic generation to derive multiple instances of the same dataset and evaluate models based on a distribution of performance measures rather than a single performance metric.
>
> Thank you again for your feedback and for facilitating a transparent discussion on these topics.
>
>
> Best Regards,
> GSM-Symbolic Authors
>
>
>
> References:
> [1] https://huggingface.co/datasets/math-ai/TemplateGSM
>
> [2] https://github.com/iiis-ai/TemplateMath

---

> > ### Public Comment · ~Yifan_Zhang16 · 2024-11-16
> > **Response to Authors' Comment on GSM-Symbolic vs TemplateGSM**
> >
> > Thank you for your detailed response and for addressing the concerns regarding the potential similarity between GSM-Symbolic and TemplateGSM. I would like to clarify and provide evidence that TemplateGSM, including its methodology, templates, and examples, has been publicly available for nearly a year, well before the timeline of your research.
> >
> > 1. **Public Availability of TemplateGSM Methodology:**
> >    - The methodology and templates used for TemplateGSM have been publicly accessible since **February 2024**, as documented on:
> >      - **GitHub**: The repository includes clear examples of problem generation using GSM8K's test and train sets:
> >        - [Templates for the test set](https://github.com/yifanzhang-pro/syntax-semantics/blob/main/template-gsm/test/gsm-test-0005-1.py).
> >        - [Templates for the train set](https://github.com/yifanzhang-pro/syntax-semantics/blob/main/template-gsm/gsm-0000-1.py).
> >      - **Prompt Design**: The specific prompts for generating templates with GPT are also available: [Prompt for Template Generation](https://github.com/yifanzhang-pro/syntax-semantics/blob/main/template-gsm/prompt.md).
> >      - **Hugging Face Dataset**: TemplateGSM, including millions of generated problems, has been hosted on Hugging Face: [TemplateGSM Dataset](https://huggingface.co/datasets/math-ai/TemplateGSM).
> >    - GitHub’s version history verifies these uploads, which conclusively establishes their early public availability.
> >
> > 2. **Distinction Between PDF Upload and Methodology Release:**
> >    - It is true that the PDF report summarizing TemplateGSM’s methodology was uploaded recently as part of an update to centralize documentation. However, this does not conflict with the fact that the methodology itself—including templates, prompts, and data—has been publicly accessible since early 2024.
> >    - The PDF serves only as a consolidated explanation of the work, which has been available for months through the resources listed above.
> >
> > 3. **Overlap in Methodology:**
> >    - GSM-Symbolic’s reliance on GSM8K templates and symbolic variations is conceptually aligned with TemplateGSM. For example, TemplateGSM similarly explores variations of mathematical problems derived from GSM8K by modifying names, numerical values, and problem structures.
> >    - The symbolic template-based generation you describe in GSM-Symbolic closely parallels the foundational approach of TemplateGSM.
> >
> > 4. **Acknowledgment and Transparency:**
> >    - While GSM-Symbolic introduces valuable new benchmarks, such as GSM-NoOp and GSM-P2, the foundational similarity in template-based symbolic generation should be acknowledged. Doing so fosters collaboration and strengthens the transparency of contributions within the research community.
> >    - The broader concept of syntactic and symbolic data generation, as you note, is indeed well-established. However, the specific implementation details and methodology of TemplateGSM have been publicly available for months and merit recognition.
> >
> > I appreciate your willingness to engage in this discussion and encourage a continued transparent dialogue to ensure fair acknowledgment of contributions. Please feel free to reach out if additional clarifications or evidence would help in addressing these points.
> >
> > Best regards,

---

> > ### Public Comment · ~Yifan_Zhang16 · 2024-11-16
> > **Follow-Up Question on GSM-Symbolic’s Template and Data Accessibility**
> >
> > Thank you for your detailed response and the clarifications provided. Since the **templates, prompts, and data of TemplateGSM** have been publicly available for 10 months through GitHub and Hugging Face, as documented below:
> >
> > - **GitHub**:
> >   - [Templates for the test set](https://github.com/yifanzhang-pro/syntax-semantics/blob/main/template-gsm/test/gsm-test-0005-1.py): Includes examples of problem generation using GSM8K’s test set.
> >   - [Templates for the train set](https://github.com/yifanzhang-pro/syntax-semantics/blob/main/template-gsm/gsm-0000-1.py): Includes examples for the GSM8K train set.
> >
> > - **Prompt Design**:
> >   - The specific prompts for generating templates with GPT are available [here](https://github.com/yifanzhang-pro/syntax-semantics/blob/main/template-gsm/prompt.md).
> >
> > - **Hugging Face Dataset**:
> >   - TemplateGSM, featuring millions of generated problems, is hosted on Hugging Face: [TemplateGSM Dataset](https://huggingface.co/datasets/math-ai/TemplateGSM).
> >
> > GitHub’s version history verifies these uploads, conclusively establishing their early public availability.
> >
> > I would like to ask:
> > - **When will GSM-Symbolic’s templates and data be made publicly available to the research community?**
> >
> > Open access to your templates and data would enable others to replicate your findings, benchmark against your work, and contribute further to this important area of research. Transparency and accessibility are crucial for fostering collaboration and advancing shared goals in evaluating mathematical reasoning capabilities in large language models.
> >
> > I look forward to your response on this matter and appreciate your engagement in this discussion.
> >
> > Best regards,

---

> ### Public Comment · ~Yifan_Zhang16 · 2024-11-16
> **Follow-Up Questions on TemplateGSM and GSM-Symbolic**
>
> Thank you for your detailed response and for engaging in this important discussion.
>
> Given that the authors of GSM-Symbolic consider their work to be **distinct** from TemplateGSM, I would like to ask for further clarification and your perspective on several related matters:
>
> 1. **Publishing Considerations:**
>    - In your view, how should these two projects—GSM-Symbolic and TemplateGSM—be positioned within the research landscape if both were submitted to AI conferences?
>    - Specifically, do you believe the authors of TemplateGSM could still submit their work to prominent AI conferences, considering the current overlaps and differences you outlined?
>
> 2. **Broader Community Input:**
>    - What are the thoughts of other reviewers and the community on this matter? It would be valuable to gather perspectives on whether projects like TemplateGSM should be recognized as distinct contributions or if their originality might be questioned given GSM-Symbolic's publication.
>
> 3. **GSM-Symbolic Authors’ Review Opinion:**
>    - If you, as the authors of GSM-Symbolic, were tasked with reviewing a submission of TemplateGSM to an AI conference, what comments or feedback would you provide?
>
> We believe these questions are critical for fostering a transparent and fair discussion in the research community. To ensure that both projects are appropriately positioned and acknowledged, we sincerely request your suggestions on how these issues can be handled equitably.
>
> Your guidance would be instrumental in clarifying the path forward, especially for researchers aiming to contribute to the advancement of mathematical reasoning in large language models.
>
> Thank you again for your time and thoughtful engagement.
>
> Best regards,

---

### Public Comment · ~Desi_R._Ivanova1 · 2024-11-21
**Statistical analysis?**

Firstly, I'd like to thank the authors for their interesting work.

A few comments and questions:

1. The work lacks statistical evaluations. Given the focus of this work to look at and evaluate distributional properties of performance, the empirical analysis completely lacks rigorous statistical evaluations. The only attempt at recognising the existence of statistical noise is when a hand-wavy "one standard deviation" criterion is mentioned in section 4.1 of the paper.

2. I wonder why the variation of performance is unexpected ("It is noteworthy that this variation even occurs").

3. What checks have been performed to ensure that GSM8K and GSM-Symbolic come from the same distribution? There is evidence of distribution mismatch between the two -- the proposed template in Figure 1 cannot generate the original question.

Finally, directly related to the discussion here:

> Additionally, modern state-of-the-art models handle operations involving 3-4 digit numbers quite effectively because of their memorization capabilities.

This is actually not the case, at least not for all models; particularly the smaller ones (which is what is evaluated in this work). You can see a plot the average success rate of correctly adding two $d$-digit numbers for Llama3-8b-instruct and Phi3.5-mini-instruct here https://substackcdn.com/image/fetch/f_auto,q_auto:good,fl_progressive:steep/https%3A%2F%2Fsubstack-post-media.s3.amazonaws.com%2Fpublic%2Fimages%2Fc69bcead-4865-4632-b482-080f732c2d31_5000x3500.png. Even Llama3-8B cannot add 4 digit numbers perfectly!

In a blog post that I won't link here for the purpose of preserving anonymity, I show (among other things) that for Phi3.5-mini-instruct, using the proposed ranges in Figure 1 vs smaller ranges that would better represent the original question, can give raise to a performance gap of ~4.5% for that one question (table with results can be found here https://www.datawrapper.de/_/klEuv/). This again links back to potential distribution mismatch between GSM8K and GSM-Symbolic, and how much of the (alleged) performance decline is due to that.

---

### Public Comment · ~Cheryl_Lee1 · 2025-02-18
**Could you please release the Noop dataset?**

As stated in the title. Since the drop on the NoOP dataset is the most significant, it may be the most important dataset, but the current GitHub repo does not contain this dataset.

---

> ### Public Comment · ~Desi_R._Ivanova1 · 2025-07-03
>
> Indeed. What's worse is that it makes it impossible to verify their results as well.

---

### Meta-Review · Area_Chair_Zmkb · 2024-12-21

**Metareview:**

The paper introduces GSM-Symbolic, a novel benchmark designed to evaluate the mathematical reasoning capabilities of large language models (LLMs) with a focus on controlled and nuanced assessments. Building on the widely used GSM8K benchmark, GSM-Symbolic employs symbolic templates to generate diverse question variants, allowing for more reliable and comprehensive evaluations of LLMs' reasoning abilities. The authors conduct extensive experiments across multiple state-of-the-art models, revealing significant performance variance when numerical values and question complexity are altered. Notably, the study shows that adding irrelevant clauses can lead to substantial performance drops, suggesting that current LLMs struggle with genuine logical reasoning and often rely on pattern-matching from training data. The paper also discusses the fragility of LLMs' mathematical reasoning, particularly as question complexity increases, and highlights potential areas for future research.

#### Contribution
1. **GSM-Symbolic Benchmark**: The introduction of GSM-Symbolic, with its symbolic templates and various question variants (e.g., GSM-names, GSM-vars, GSM-NoOp), provides a valuable tool for assessing LLMs' robustness and reasoning capabilities in mathematical contexts.
2. **Empirical Insights**: The paper offers empirical evidence of LLMs' limitations, demonstrating performance variance and fragility when faced with slight modifications or increased complexity in mathematical problems.
3. **Discussion on Reasoning**: The study contributes to the ongoing discourse about what constitutes "true logical reasoning" in LLMs, proposing a heuristic based on performance consistency across different instantiations of problems.
4. **Broad Evaluation**: The authors evaluate a large number of models, providing a comprehensive overview of current state-of-the-art methods and their limitations.

#### Weaknesses
1. **Statistical Rigor**: The paper lacks rigorous statistical evaluations, particularly in assessing the significance of performance variations across different models and question variants.
2. **Novelty Concerns**: Some reviewers argue that the methods used (e.g., altering numerical values, adding irrelevant information) are not novel and have been explored in prior works, raising questions about the paper's originality.
3. **Ambiguity in Reasoning Definition**: The paper uses terms like "true logical reasoning" and "genuine mathematical reasoning" without providing clear definitions, leading to potential confusion about the study's claims.
4. **Limited Exploration of Computational Complexity**: The impact of computational complexity on model performance is not thoroughly addressed, potentially confounding the interpretation of reasoning limitations.
5. **Potential Overlap with Existing Datasets**: Concerns were raised about the similarity between GSM-Symbolic and the TemplateGSM dataset, although the authors argue that their work was developed independently.

**Additional Comments On Reviewer Discussion:**

1. **Statistical Rigor (Desi R. Ivanova, Yt9o):**
   - **Concern**: The paper lacks statistical evaluations, particularly in assessing performance variations.
   - **Response**: The authors added statistical significance results based on one-sample t-tests in Appendix A.3 (Figure 10), acknowledging the complexity of statistical analysis and committing to further investigation.

2. **Novelty Concerns (FK55):**
   - **Concern**: The methods used are not novel and have been explored in prior works.
   - **Response**: The authors clarified that their primary contribution lies in providing a nuanced analysis of performance distributions across different setups, rather than merely introducing a new dataset. They emphasized the significance of understanding performance variance and degradation patterns.

3. **Ambiguity in Reasoning Definition (FK55, 7pZc):**
   - **Concern**: The paper uses ambiguous terms for reasoning without clear definitions.
   - **Response**: The authors proposed a definition of "logical reasoning" based on the ability to solve novel problems through logical steps, distinguishing it from memorization. They acknowledged the ongoing challenge in the field to define reasoning rigorously and welcomed further discussion.

4. **Computational Complexity (7pZc):**
   - **Concern**: The impact of computational complexity on model performance was not adequately addressed.
   - **Response**: The authors provided an analysis of arithmetic accuracy across different benchmarks, demonstrating high accuracy in both simple and complex scenarios. They argued that the observed performance drops are primarily due to reasoning limitations rather than computational errors.

5. **Potential Overlap with TemplateGSM (Yifan Zhang):**
   - **Concern**: Potential overlap with the TemplateGSM dataset, raising questions about originality.
   - **Response**: The authors clarified that GSM-Symbolic was developed independently and highlighted differences in methodology and goals. They emphasized that their work focuses on evaluating models rather than training them, using unique templates and a broader set of benchmarks.

6. **Performance Variance and Model Robustness (Yt9o, wG4b):**
   - **Concern**: Inconsistencies in model performance across different setups, particularly with models like GPT-4o showing robustness.
   - **Response**: The authors explained that performance variance is expected and that even robust models show limitations in more challenging setups (e.g., GSM-NoOp). They also clarified the use of greedy decoding to ensure consistency in model outputs, addressing concerns about variance due to sampling strategies.

7. **Clauses in GSM-NoOp (wG4b):**
   - **Concern**: The criteria for adding clauses in GSM-NoOp were unclear and potentially subjective.
   - **Response**: The authors explained that clauses were added by an expert human based on heuristics designed to be inconsequential to the problem's logic, ensuring consistency across models without bias.

The authors demonstrated a strong commitment to addressing reviewer concerns, significantly improving the paper through additional analyses, clarifications, and a more comprehensive discussion of findings. While some limitations, such as the definition of reasoning and potential novelty concerns, remain, the paper's contributions to understanding LLMs' mathematical reasoning capabilities are substantial. The introduction of the GSM-Symbolic benchmark, along with the detailed empirical analysis, positions this work as a valuable resource for the research community.

Although there are still two negative reviews, I still lean toward accept this paper, but the authors need not to over-claim this paper (as many AI media coverage) saying LLM cannot reason. As results show most sota LLMs only have little performance drop. This dataset is useful to detect those LLMs that overfit to GSM8k dataset. Authors shall also include some recent reasoning models like o1, and qwq.

---

> ### Public Comment · ~Desi_R._Ivanova1 · 2025-07-02
> **Make Peer Review Great Again**
>
> Dear AC,
>
> Thank you for summarising the contributions, weaknesses, key points raised during the discussion period, along with the promises the authors made for correction and further investigation.
>
> I only recently had a chance to look at the revised paper, specifically the promised statistical analysis. I'm afraid the revision made **did not constitute a meaningful correction but a superficial gesture towards rigour at best, and severe negligence at worst**. I'm including detailed notes below and looking forward to hearing what options are available for post-review correction or escalation.
>
> Kind regards,
>
> Desi R Ivanova
>
> ---
> A per your summary:
> > **Statistical Rigor** (Desi R. Ivanova, Yt9o):
> **Concern**: The paper lacks statistical evaluations, particularly in assessing performance variations.
> **Response**: The authors added statistical significance results based on one-sample t-tests in Appendix A.3 (Figure 10), acknowledging the complexity of statistical analysis and committing to further investigation
>
> Looking at the revised paper, the authors indeed added a one-sample t-test in one of the charts in the Appendix. Unfortunately, both the test they used and the way it was conducted seem to be incorrect.
>
> ## On the correctness of the test
>
> 1. A one-sample t-test is not appropriate in this context as the evaluation involves two datasets (GSM8K and GSM-Symbolic). The authors have access to their raw data, so they could have performed a proper two-sample comparison.
>
> 2. More critically, for a t-test to be valid, the sample mean and variance [must be independent](https://en.wikipedia.org/wiki/Student%27s_t-test#Assumptions) (footnote 4 on page 18 is incomplete on this point). This assumption does not hold here because the outputs are Bernoulli variables (Binomial when aggregated), whose variance is directly determined by the mean.
>
> A two-sample z-test would have been more appropriate. I performed this analysis in this [initial quick post](https://substack.com/@desirivanova/p-150508215) (later expanded and [published in the ICLR Blog Post track](https://iclr-blogposts.github.io/2025/blog/towards-more-rigorous-llm-evals)), complete with a [publicly available spreadsheet](https://docs.google.com/spreadsheets/d/1Ul6ZgFXf_II5EFUCgnJ9hSIQYwHxogxYBmwDn_bA4sA/edit?usp=sharing) showing the calculations! These resources were publicly available to the authors throughout the review process. They could have used them or reached out directly (given I posted under my real name).
>
> ## The execution of the test
> The authors provide no description of **how** their statistical test was conducted. They state the Null hypothesis as "*50 different performance results on GSM-Symbolic differ from the original GSM8K score*" (Appendix A.3, page 18). The Null hypothesis, as the name suggests, posits there is no difference, so they must be describing the alternative. Given they are performing a one sample test, it is unclear whether they treat the GSM8K or the GSM-Symbolic score as fixed. This has huge implications as it determines what standard error is used. It is also unclear whether the test was one- or two-sided.
>
> **Having performed a test that is methodologically invalid on multiple levels**, the authors claim "*for an overwhelming majority of models (except Llama3-8B and GPT-4o), the results are statistically significant*" (Appendix A.3, page 18). This is in **direct contradiction** to my own careful analysis (see Section 4.2.2 in the [published ICLR blog post](https://iclr-blogposts.github.io/2025/blog/towards-more-rigorous-llm-evals), or the [publicly available spreadsheet](https://docs.google.com/spreadsheets/d/1Ul6ZgFXf_II5EFUCgnJ9hSIQYwHxogxYBmwDn_bA4sA/edit?usp=sharing) accompanying my initial analysis), where I found that only 3 models showed significant decrease in performance (Gemma-7b, Mistral-7b-instruct-v0.1, Phi-2) and 1 showed significant increase (Llama3-8b).
>
> ## Other issues
> The promise of "further investigation" appears to have remained just that - a promise.
>
> Needless to say, there is quite a lot more analysis in the [published ICLR blog post](https://iclr-blogposts.github.io/2025/blog/towards-more-rigorous-llm-evals). For convenience, here are what I believe to be the other key technical issues:
> - Over-interprets and over-sensationalises expected statistical variations.
> - Completely ignores alternative explanations: lack of reasoning is a plausible explanation, but so is distribution mismatch between the original GSM8K and the Symbolic version they introduce.
> - The paper’s own data suggests this mismatch: the example template given in the paper (Figure 1) does not even reproduce the actual question in the original dataset.
>
> This, combined with what appears to be inaccurate statistical analysis of their results, suggests that the claims made are unsupported and the paper should be either corrected or retracted.
>
> Thank you again for your time and for overseeing a complex review process.

---

### Decision · Program_Chairs · 2025-01-22

Accept (Poster)